# Metabolic *N*-Dealkylation and *N*-Oxidation as Elucidators of the Role of Alkylamino Moieties in Drugs Acting at Various Receptors

**DOI:** 10.3390/molecules26071917

**Published:** 2021-03-29

**Authors:** Babiker M. EH-Haj

**Affiliations:** Department of Pharmaceutical Sciences, College of Pharmacy and Health Sciences, University of Science and Technology of Fujairah, Emirate of Fujairah, Fujairah 2022, United Arab Emirates; b.elhag@ustf.ac.ae; Tel.: +971-567-204-338

**Keywords:** *N*-alkylamino moieties, metabolic *N*-dealkylation, metabolic *N*-oxidation, pharmacologic activity, physicochemical properties, *N*-desalkylamino metabolite drugs, *N*-oxide metabolite prodrugs

## Abstract

Metabolic reactions that occur at alkylamino moieties may provide insight into the roles of these moieties when they are parts of drug molecules that act at different receptors. *N*-dealkylation of *N*,*N*-dialkylamino moieties has been associated with retaining, attenuation or loss of pharmacologic activities of metabolites compared to their parent drugs. Further, *N*-dealkylation has resulted in clinically used drugs, activation of prodrugs, change of receptor selectivity, and providing potential for developing fully-fledged drugs. While both secondary and tertiary alkylamino moieties (open chain aliphatic or heterocyclic) are metabolized by CYP450 isozymes oxidative *N*-dealkylation, only tertiary alkylamino moieties are subject to metabolic *N*-oxidation by Flavin-containing monooxygenase (FMO) to give *N*-oxide products. In this review, two aspects will be examined after surveying the metabolism of representative alkylamino-moieties-containing drugs that act at various receptors (i) the pharmacologic activities and relevant physicochemical properties (basicity and polarity) of the metabolites with respect to their parent drugs and (ii) the role of alkylamino moieties on the molecular docking of drugs in receptors. Such information is illuminative in structure-based drug design considering that fully-fledged metabolite drugs and metabolite prodrugs have been, respectively, developed from *N*-desalkyl and *N*-oxide metabolites.

## Table of Contents


**Section**

**Topics**

**Pages**


**Abstract**
1
**Section 1**
Introduction3
**Section 2**
Neurotransmitter (NT) Reuptake Inhibitors4
Section 2.1
Serotonin-Norepinephrine Reuptake Inhibitors4
Section 2.1.1
Imipramine and Amitriptyline4, 5
Section 2.1.2
Clomipramine6
Section 2.1.3
Venlafaxine6, 7
Section 2.1.4
Doxepin7
Section 2.2
Selective Norepinephrine Reuptake Inhibitors7
Section 2.2.1
Maprotiline7, 8
Section 2.2.2
Atomoxetine8
Section 2.3
Selective Serotonin Reuptake Inhibitors9
Section 2.3.1
Fluoxetine9
Section 2.3.2
Citalopram/Escitalopram9, 10
Section 2.3.3
Sertraline10
Section 2.3.4
Fenfluramine10, 11
**Section 3**
Dopaminergic, Serotonergic, Adrenergic and *N*-methyl-D-aspartate (NMDA) Receptor Blockers11
Section 3.1
Loxapine/Amoxapine11
Section 3.2
Clozapine11, 12
Section 3.3
Mianserin12
Section 3.4
Mirtazapine12, 13
Section 3.5
Olanzapine13
Section 3.6
Ketamine14
Section 3.7
Chlorpromazine14, 15
Section 3.8
Promazine15
**Section 4**
Histamine-1 Receptor Antagonists16
Section 4.1
Diphenhydramine16
Section 4.2
Azelastine16, 17
Section 4.3
Prometazine17
**Section 5**
Opioid-mu-Receptor Agonists17
Section 5.1
Morphine/Codeine17, 18
Section 5.2
Tramadol18
Section 5.3
Propoxyphene19
Section 5.4
Meperidine19
**Section 6**
Calcium-Channel Blockers20
Section 6.1
Verapamil20
Section 6.2
Diltiazem20
Section 6.3
Amiodarone21
**Section 7**
Drugs That Act at Sodiuzm Channels21
Section 7.1
Local Anesthetics21
Section 7.1.1
Lidocaine21
**Section 8**
Drugs That Act at GABAnergic Receptors22
Section 8.1
Zopiclone22
**Section 9**
Muscarinic-Receptor Blockers22
Section 9.1
Tolterodine/Fesoterodine22, 23
Section 9.2
Oxybutynin23
**Section 10**
“If” Channel Blockers23
Section 10.1
Ivabradine23, 24
**Section 11**
Drugs That Act as Enzyme Inhibitors24
Section 11.1
Sildenafil24
**Section 12**
Drugs That Act on Microorganisms24
Section 12.1
Chloroquine/Hydroxychloroquine24–26
**Section 13**
Anticancer Drugs26
Section 13.1
Imatinib26
Section 13.2
Dacarbazine26, 27
Section 13.3
Tamoxifen27
Section 13.4
Tormifene27, 28
**Section 14**
Metabolic *N*-dealkylation and *N*-oxidation28
Section 14.1
Metabolic *N*-dealkylation28, 29
Section 14.1.1
Focused *N*-dealkylation cases30
Section 14.1.1.1
Loss of Pharmacologic Activity30
Section 14.1.1.2
Modification of Receptor Inhibition Selectivity30, 31
Section 14.1.1.3
Activation of Prodrugs31
Section 14.1.1.4
Potential Drug Candidates (Metabolite Drugs)31
Section 14.2
*N*-Oxidation Of Tertiary-Alkylamino-Moiety-Containing Drug32
**Section 15**
Conclusions33

## 1. Introduction

Alkylamino moieties, either open chain aliphatic (secondary or tertiary), or heterocyclic tertiary ones, are common in drug molecules of various pharmacological classes. Their basicity and polarity are essential for drug action. They are found in antidepressants, antihistamines, narcotic analgesics, local anesthetics, as well as other drug classes. The order of prevalence of the alky groups in alkylamino moieties is methyl > ethyl > isopropyl > *tert*-butyl > others. Methyl, ethyl, and isopropyl groups are usually found in drug molecules as tertiary *N*,*N*-dimethylamino, *N*,*N*-diethylamino, or *N*,*N*-diisopropylamino moieties, respectively. In the metabolism of drug molecules containing *N*,*N*-dimethylamino, *N*,*N*-diethylamino and *N*,*N*-diisopropylamino moieties, the alkyl groups are mostly removed sequentially to give secondary and primary amino groups. On the other hand, *tert*-butyl groups are usually less prone to metabolic oxidative dealkylation (Figure 1). Intrinsic secondary *N*-alkylamino moieties–mostly methylamino–are also encountered in some drug molecules. Another route of *N*-alkylamino moiety metabolism is *N*-oxygenation, which is specific to only tertiary *N*,*N*-dialkylamino moieties either open-chain aliphatic or heterocyclic [1]. 

Mechanistically, CYP450-catalyzed *N*-dealkylation involves as a first step the hydroxylation of the carbon atom of the alkyl group that is linked to the nitrogen atom (α-carbon atom). This hydroxylated metabolite is unstable. It breaks spontaneously into two molecules: the dealkylated metabolite (e.g., an amine), and an aldehyde (e.g., formaldehyde after demethylation, acetaldehyde after deethylation, etc.) [1]. The reaction is shown for methyl and ethyl secondary amines in Figure 1, which can be aliphatic or heterocyclic (or aromatic). Similarly, tertiary amines are dealkylated in a similar way by consecutive hydroxylation of the alkyl groups at the carbon that is linked to the nitrogen atom [1]. 

Depending on their class, alkylamino moieties interact with receptors or enzymes via hydrogen bonding, ion-dipole, ion-ion and van der Waals bindings as depicted in Figure 2 [2,3].

The alkylamino-moieties-containing drugs cited in this review may be categorized according to the receptors upon which they act.

## 2. Neurotransmitter Reuptake Inhibitors

The drugs that contain alkylamino moieties in this class belong to four categories: (i) serotonin-norepinephrine reuptake inhibitors (SNRI), (ii) selective norepinephrine reuptake inhibitors (NRI), (iii) selective serotonin reuptake inhibitors (SRI), and (iv) serotonin-norepinephrine-dopamine reuptake inhibitors (SNDRI).

### 2.1. Serotonin-Norepinephrine Reuptake Inhibitors (SNRI)

To this subclass belong the tricyclic antidepressants imipramine and amitriptyline, which are SNRI with preference for SRI, are respectively metabolized by *N*-demethylation to desmipramine and nortriptyline, (Figure 3 and Figure 4, respectively). Desipramine and nortriptyline have been developed into drugs of their own rights; they have preference for NR inhibition over SR inhibition. 

#### 2.1.1. Imipramine and Amitriptyline

Both imipramine and amitriptyline (Figure 3 and Figure 4, respectively) are aliphatic tertiary-amine tricyclic antidepressants. The two drugs are metabolized by *N*-demethylation to equiactive secondary-amine metabolites, desmipramine and nortriptyline, respectively, as shown in Figure 3 and Figure 4 [4,5,6,7,8]. The latter two drugs are further metabolized by *N*-demethylation to inactive primary amine metabolites [9]. Hydroxylation is also an important metabolic pathway of the four drugs (Figure 3 and Figure 4).

The four tricyclic antidepressant drugs are SNRI with imipramine and amitriptyline being more active as SRIs than NRIs and desmipramine and nortriptyline being more active as NRIs than SRIs [10]. Another metabolic pathway of imipramine and amitriptyline is *N*-oxygenation resulting in the formation of imipramine-*N*-oxide and amitriptyline-*N*-oxide, respectively. The two *N*-oxide metabolites have been developed as prodrugs of imipramine and amitriptyline since they (the *N*-oxide metabolites) are bioreduced in vivo to the tertiary amine parent drugs, imipramine and amitriptyline [11].

Reporting on comparison between imipramine and desmipramine, Rose and Westhead [12] found no difference between the two drugs in patients with primary depression regarding antidepressive effect or onset of action. According to the authors [12], reactive and endogenous depression responded equally well to either drug.

The pKa and log *p* values of the imipramine and desmipramine are given in Figure 3; the pKa and log *p* values of amitriptyline and nortriptyline are given in Figure 4.

#### 2.1.2. Clomipramine

Clomipramine (Figure 5), a 3-chloro analog of imipramine, is a dibenzazepine-derivative tricyclic antidepressant (TCA). It contains a dimethylamino propyl moiety. Clomipramine acts on both noradrenergic and serotonergic transporters; however, with selectively for the serotonin transporter by inhibiting transporter action at presynaptic neuronal sites [13]. Inhibition of the serotonin transmitter by clomipramine is in contrast to its principal active metabolite, *N*-desmethylclomipramine (Figure 5), which principally acts as antagonist of noradrenergic transporter receptor [13,14]. The effectiveness of clomipramine, compared to other TCAs, in the management of obsessive-compulsive disorder (OCD) may be related to its relative specificity for serotonin reuptake system inhibition [14]. This observation may suggest that OCD might be caused, in part, by dysregulation of the serotonergic system. Further metabolic pathways of clomipramine include aromatic ring hydroxylation to active 8-hydroxyclomipramineand and *N*-oxygenation to clomipramine-*N*-oxide [15]. The pKa and log *p* values of clomipramine and *N*-desmethylclomipramine are given in Figure 5.

#### 2.1.3. Venlafaxine

Venlafaxine (Figure 6) is an antidepressant drug that acts as both serotonin and norepinephrine reuptake inhibitor. It contains a dimethylaminomethyl moiety. The metabolic pathways of venlafaxine are depicted in Figure 6 [16,17,18,19]. While *O*-desmethylvenlafaxine is equiactive and equipotent with the parent drug as antidepressant and has been developed into a drug of its own right under the name of desvenlafaxine, *N*-desmethylvenlafaxine is devoid of antidepressant activity [20,21]. The pKa and log *p* values of venlafaxine and desvenlafaxine are given in Figure 6.

#### 2.1.4. Doxepin

Doxepin (Figure 7) is a tricyclic antidepressant of the dibenzoxepine class. It contains a dimethylaminopropylidino moiety and exists in two geometric forms: *E* and *Z* in the ratio of 85:15, respectively [22]. The *Z* isomer is more active than the *E* isomer as antidepressant [23]. As far as the mechanism of action of doxepin is concerned, the *E* isomer is NRI while the *Z* isomer is SRI [24,25]. However, both isomers are metabolized by *N*-demethylation to give *E*-nordoxepin and *Z*-nordoxepin, which are active antidepressants and by *N*-oxidation to give *E*-doxepin-*N*-oxide and *Z*-doxepin-*N*-oxide, which are inactive as antidepressants [26]. The pKa and log *p* values of *E*-doxepin and *E*-N-nordoxepin are given in Figure 7.

### 2.2. Selective Norepinephrine Reuptake Inhibitors (NRIs)

#### 2.2.1. Maprotiline

Maprotiline (Figure 8) is a tetracyclic antidepressant. It contains a secondary methylaminopropyl moiety. The mechanism of action of maprotiline involves selective norepinephrine neuronal reuptake inhibition. The metabolic pathways of maprotiline are depicted in Figure 8 with *N*-desmethylmaprotiline forming the major active metabolite [27,28,29,30,31]. The log *p* values of maprotiline and *N*-desmethylmaprotiline in addition to the pKa value of maprotiline are given in Figure 8. No value has been found for the pKa value of *N*-desmethylmaprotiline; however, it can be estimated to be higher than that of maprotiline.

#### 2.2.2. Atomoxetine

Atomoxetine (Figure 9) is a selective NRI used to treat attention deficit hyperactivity disorder (ADHD). It contains a secondary ethylmethylamino moiety and is metabolized as per the pathways shown in Figure 9 [32,33,34,35,36,37,38]. While aromatic-ring hydroxylation does not affect the blockade of the NET and produces an equipotent metabolite to the parent drug [35], *N*-demethylation causes nearly 20-fold loss of pharmacologic activity with respect to atomoxetine [37]. The pKa and log *p* values of atomoxetine and 4-hyroxy-*N*-desmethylatomoxetine and the pKa value of *N*-desmethylatomoxetine are given in Figure 9. No log *p* value has been found for *N*-desmethylatomoxetine; however, it can be estimated to be lower than that of atomoxetine.

### 2.3. Selective Serotonin Reuptake Inhibitors (SSRI)

#### 2.3.1. Fluoxetine

In contrast to TCAs, fluoxetine (Figure 10) is a selective serotonin reuptake inhibitor (SSRI). It is used for the treatment of depression, bulimia nervosa and obsessive-compulsive disorder (OCD) [39].

Structurally, fluoxetine is characterized by the presence of a methylaminopropyl group, as depicted in Figure 8. Fluoxetine is extensively metabolized in the liver. The only identified active metabolite, norfluoxetine, is formed by *N*-demethylation of fluoxetine [39,40,41,42,43]. Fluoxetine is a racemic mixture of two enantiomers: *R* and *S*-fluoxetine. *S*-fluoxetine is slightly more potent in the inhibition of serotonin reuptake than *R*-fluoxetine. The difference is, however, much more pronounced for the active metabolite *S*-norfluoxetine, which has about twenty-fold higher serotonin-reuptake blocking potency than the *R*-norfluoxetine [43]. The four compounds (*R*- and *S*-fluoxetine and their corresponding metabolites) differ also in their kinetics. After several weeks of treatment, the plasma concentration of both *S*-enantiomers is about two times higher than the concentration of the *R*-enantiomers [43].

Fluoxetine has now largely replaced older and less safe drugs such as tricyclic antidepressants. Different cytochrome P450 isoforms are involved in the metabolism of fluoxetine, however, the main active metabolite, norfluoxetine, is produced by the CYP2D6 action in the human liver [39,40,41,42,43]. The pKa and log *p* values of fluoxetine, norfluoxetine and norfluoxetine glucuronide are given in Figure 10.

#### 2.3.2. Citalopram/Escitalopram

Citalopram (Figure 11) is a SSRI. It is used to treat depression for panic attacks. Citalopram is a chiral drug [44]. The substantially more active *S*-enantiomer has been developed into a drug of its own right under the name of escitalopram. As shown in Figure 11, citalopram contains a dimethylaminopropyl moiety and is primarily sequentially metabolized by oxidative *N*-demethylation to *N*-demethylcitalopram (DCT) by CYP3A4 and to *N*,*N*-didemethylcitalopram (DDCT) by CYP2D6 [45,46,47]. Other metabolites include inactive citalopram-*N*-oxide and a deaminated propionic acid derivative (Figure 11). In humans, unchanged citalopram is the predominant compound in plasma [47]. At steady state, the concentrations of citalopram’s metabolites, DCT and DDCT, in plasma are approximately one-half and one-tenth, respectively, that of the parent drug [47].

In vitro studies show that citalopram is at least 8 times more potent than its metabolites in inhibiting serotonin reuptake [47], suggesting that the metabolites evaluated do not likely contribute significantly to the antidepressant actions of citalopram.

The pKa and log *p* values of citalopram and norcitalopram are given in Figure 11.

#### 2.3.3. Sertraline

Sertraline (Figure 12) is an antidepressant of the selective serotonin reuptake inhibitor (SSRI) class. It is primarily prescribed for major depressive disorder (MDD) in adult outpatients as well as obsessive-compulsive disorder (OCD), panic disorder, and social anxiety disorder, in both adults and children. In 2013, it was the most prescribed antidepressant and second most prescribed psychiatric medication (after alprazolam) on the U.S. retail market, with over 41 million prescriptions annual in 2013 [48]. Sertraline contains a methylamino moiety and is metabolized by *N*-demethylation to *N*-desmethylsertraline [48,49,50]. The metabolite is 5 to 10 times less potent as SSRI than the parent drug and accordingly its clinical contribution is negligible [48]. The pKa and log *p* values of sertraline and norsertraline are given in Figure 12.

#### 2.3.4. Fenfluramine

Fenfluramine (Figure 13) is a serotonin reuptake inhibitor that also acts by causing release of 5-HT from stores [51,52]. It was as appetite inhibitor before being withdrawn; it has, however, been reinstated in the treatment of Dravet syndrome (a type of epileptic disease) [52]. Fenfluramine contains an aliphatic secondary ethylaminopropyl moiety and is metabolized to the main active product, *N*-desmethylfenfluramine (norfenfluramine) (Figure 13) [53,54]. The pKa and log *p* values of fenfluramine and norfenfluramine are given in Figure 13.

## 3. Dopaminergic, Serotonergic, Adrenergic and *N*-methyl-d-aspartate (NMDA) Receptor Blockers

### 3.1. Loxapine/Amoxapine

Loxapine (Figure 14) is a neuroleptic of the dibenzoxazepine class. It is mainly a dopamine antagonist, but also a serotonin 5-HT2 blocker, used in the management of schizophrenia [55]. It contains a tertiary heterocyclic methylamino group and is metabolized in vivo to *N*-desmethylloxapine and 8-hydroxyloxapine [56,57,58], two compounds with antidepressant activity; however, only desmethylloxapine has shown favorable pharmacodynamic, pharmacokinetic and toxicological properties to be developed into a fully-fledged drug under the name of amoxapine. The pKa and log *p* values of loxapine and *N*-desmethylloxapine (amoxapine) are given in Figure 14.

### 3.2. Clozapine

Clozapine (Figure 15) is a dibezodiazepine atypical neuroleptic antipsychotic agent used in the treatment of schizophrenia. It acts as an antagonist of dopamine and serotonin receptors [59]. It contains a heterocyclic tertiary methylamino moiety and is metabolized in humans to *N*-desmethylclozapine, which has limited antipsychotic activity and clozapine-*N*-oxide, which is inactive (Figure 15) [60,61,62,63]. The pKa and log *p* values of clozapine and *N*-desmethylclozapine are given in Figure 15.

### 3.3. Mianserin

Mianserin (Figure 16) is tetracyclic second-generation typical antidepressant used in the treatment of depression. It mainly acts as a serotonin-receptor antagonist and to a lesser extent as norepinephrine antagonist [64]. Mianserin contains a heterocyclic tertiary methylamino moiety and is metabolized by *N*-demethylation, aromatic-ring hydroxylation and *N*-oxygenation as depicted in Figure 16 [65,66,67,68]. The blocking of the two receptors is shared to a lesser extent by *N*-desmethylmianserin and 8-hyroxymianserin. The *N*-oxide metabolite is inactive [66]. The pKa and log *p* values of mianserin and *N*-desmethylmianserin are given in Figure 16.

### 3.4. Mirtazapine

Mirtazapine (Figure 17) is a pyrazinopyridobenazepine that acts as atypical antidepressant through two mechanisms: it antagonizing 5-HT2 and 5-HT3 receptors as well as it increases noradrenaline release into the synapse [69]. Mirtazapine contains a tertiary heterocyclic methylamino moiety. As shown in Figure 17, mirtazapine is metabolized by *N*-demethylation to *N*-desmethylmirtazapine, aromatic-ring hydroxylation at position 8 to 8-hydrpxymirtazapine and *N*-oxidation to mirtazapine-*N*-oxide [70,71]. The first two metabolites have much lower antidepressant activity than mirtazapine while the *N*-oxide metabolite is inactive [70]. Further, mirtazapine is a chiral drug as indicated in Figure 18. The levo enantiomer of mirtazapine has a two-fold elimination half-life longer than the dextro enantiomer [70]. The levo enantiomer, therefore, achieves plasma levels that are about 3 times as high as that of the dextro enantiomer [70]. The pKa and log *p* values of mirtazapine and *N*-desmethylmirtazapine are given in Figure 17.

### 3.5. Olanzapine

Olanzapine (Figure 18) is a second-generation antipsychotic used in the treatment of schizophrenia, bipolar disorder [ref] and for treatment-resistant depression. The mechanism of action of olanzapine in the management of schizophrenia has been proposed as mediation through a combination of dopamine and serotonin type 2 (5HT2) antagonisms [72]. Olanzapine contains a heterocyclic tertiary methylamino moiety and is metabolized as shown in Figure 18 to active 2 and 7-hydroxy derivatives, *N*-demethylation to 4′-*N*-desmethylolanzapine, glucuronidation at position 10 to olanzapine-10-glucuronide and 4′-*N*-oxygenation to olanzapine-oxide [72,73,74,75,76]. The latter three metabolites are reported to lack pharmacologic activity at the observed concentrations [72]. The pKa and log *p* values of olanzapine and *N*-desmethylolanzapine are given in Figure 18.

### 3.6. Ketamine

Ketamine (Figure 19) is an *N*-methyl-D-aspartate (NMDA) receptor antagonist with a potent anesthetic effect [77]. Ketamine is a chiral drug and exists as *R* and *S* enantiomers; the *S*-enantiomer is marketed under the name of esketamine for use as an anesthetic [78,79]. Ketamine contains a methylamino moiety bonded to a cyclohexanone moiety. It is mainly metabolized to active *N*-desmethylketamine (norketamine), which is further metabolized by cyclohexanone hydroxylation at positions 4, 5 and 6 of the cyclohexanone ring as depicted in Figure 19 [78,79,80,81]. All the hydroxylated norketamine metabolites are inactive and are further metabolized in phase II to inactive glucuronide conjugates at the hydroxyl (OH) groups [79]. 5-Hydroxynorketamine is further metabolized by dehydrogenation to 5,6-dehydronorketamine, which is and active anesthetic and has proved to be of forensic significance because of its long half-life [79].

The pKa and log *p* values of ketamine and norketamine are given in Figure 19. It is to be noted that the pKa of ketamine corresponding to the secondary methylamino moiety is higher than that of norketamine, which corresponds to the primary amino group (Figure 19). This is explained by the positive inductive effect of the methyl group increasing the electron density on the nitrogen with the consequent increase in basicity.

### 3.7. Chlorpromazine

Chlorpromazine (Figure 20) is the prototype of the phenothiazine class of antipsychotics/neuroleptics. It produces its antipsychotic effect by the post-synaptic blockade at the dopamine D2 receptors in the mesolimbic pathway of the brain [82,83]. Due to its interaction with several sites including histaminergic, cholinergic, adrenergic, and serotonergic receptors, chlorpromazine has indications as antiemetic, major tranquilizer and in the treatment of intractable hiccups, in addition to the side effects associated with those interactions.. Chlorpromazine contains a dimethylaminopropylene moiety; it is mainly metabolized through the pathways depicted in Figure 20 [84,85,86,87,88]. The pharmacologically active metabolites of chlorpromazine include promazine, *N*-desmethylchlorpromazine, and 7-hydroxychlorpromazines [87]. The 5-sulfoxide and *N*-oxide metabolites (Figure 20) are pharmacologically inactive [87]. Further, the metabolism of chlorpromazine involves the 7-hydroxylation and 5-sulfoxidation of *N*-desmethylchlorpromazine [87]. The 7-hydroxy and *N*-desmethyl metabolites also form glucuronide conjugates in phase II [87].

The pKa and log *p* values of chlorpromazine are given in Figure 20. No corresponding values have been found for *N*-desmethylchlorpromazine; however, the values can be estimated as being, respectively, higher and lower than those of chlorpromazine.

### 3.8. Promazine

Promazine (Figure 21) belongs to the phenothiazine class of antipsychotic/neuroleptic class of drugs that act at the D2 dopamine receptor in the mesolimbic pathway of the brain [82]. It is used in the short-term treatment of disturbed behavior. Due to its interaction with the histamine-1 receptor, it is also used as antiemetic [89]. Promazine contains a dimethylaminopropylene moiety and is mainly metabolized via the routes shown in Figure 21 [88,89,90,91,92,93,94,95]. Despite a lack of literature reports on the pharmacologic activity of promazine metabolites, predictions can be made with reference to known cases: the *N*-desmethyl and 7-hydroxymetabolites have attenuated activities; the 5-sulfoxide metabolite is devoid of activity. In a detailed study of promazine metabolism, Goldenberg et al. (1964) [91] reported the formation of 3-hydroxy-*N*-desmethylpromazine, 5-sulfoxide-*N*-desmethylpromazine and glucuronide and sulfate conjugation of 3-hydroxypromazines (Figure 21). The pKa and log *p* values of promazine and *N*-desmethylpromazine are given in Figure 21.

## 4. Histamine-1 Receptor Inverse Antagonists

The first-generation H1-anithistamines are characterized by the presence of a diaryl-ring system and a dimethylamino moiety bridged by a 2-3 carbon chain. The protonated amino group and the diaryl-ring system represent the primary pharmacophores in the first-generation H1-antihistamines [96]. Having a pKa of ~9, the amino group interacts with the H1-histamine receptor via ion-ionic or hydrogen bond bindings while the diaryl-ring system interacts with the receptor via hydrophobic binding [96]. As depicted in Figure 2, the protonated amino nitrogen provides the ion in the H1-antihistamine while the receptor provides the aspartate amino-acid residue that contains the negatively charged carboxylate group (COO^−^) needed for the ion-ion binding interaction. Furthermore, the receptor provides the hydrogen-containing groups involved in hydrogen bonding, mostly the OH group in serine or glutamine.

### 4.1. Diphenhydramine

Diphenhydramine (Figure 22), of the ethanolamine chemical class, is taken to represent the first-generation H1-antinistamines. Diphenhydramine is metabolized as per the pathways shown in Figure 22 [96,97]. According to Foye (2013) [98], “*N*-desmethyl and *N*,*N*-didesmethyl metabolites contribute very little to the observed antihistaminic properties of diphenhydramine’’. On the other hand, the acetamide and the carboxylic-acid metabolites (Figure 22) lack the pharmacophoric amino group and are therefore devoid of H1-antihistaminic activity. The pKa and log *p* values of diphenhydramine and *N*-desmethyldiphenhydramine are given in Figure 22.

### 4.2. Azelastine

Azelastine (Figure 23), a phthalazine derivative, is a second-generationH1-antihistamine and mast cell stabilizer. It contains a heterocyclic tertiary methylamino moiety and is metabolized by oxidative *N*-demethylation by the cytochrome P450 enzymes to the principal active product *N*-desmethylazelastine, which has H1-receptor antagonistic activity with longer duration of action than the parent drug [99,100,101,102,103]. The pKa and log *p* values of azelastine and *N*-desmethylazelastine are given in Figure 23.

### 4.3. Promethazine

Promethazine (Figure 24) is a phenothiazine derivative. The introduction of a methyl branch in the alkyl chain of antipsychotic phenothiazines, such as promazine and chlorpromazine (Figure 24), has introduced a detour in the mechanism of action of promethazine [104]. With such structural modification, promethazine belongs to the tricyclic H1-antihistamines and is used to treat allergic reactions as well as nausea and emesis. Promethazine contains a dimethylaminoisopropyl moiety and is metabolized by CYP2D6 isozyme via the pathways shown in Figure 24 [104,105,106,107,108,109,110]. The pKa and log *p* values of promethazine and *N*-desmethylpromethazine are given in Figure 24.

## 5. Opioid-mu Receptor Agonists

Alkylamino moieties (open chain aliphatic or heterocyclic) feature in some mu-receptor agonists; they are metabolized by oxidative dealkylation by CYP450 isozymes to *N*-desalkylamino products.

### 5.1. Morphine and Codeine

A heterocyclic tertiary methylamino moiety is a common structural feature of morphine and codeine (Figure 25) and other semisynthetic opiate narcotic analgesics. In morphine and codeine, this group is subject to metabolic *N*-demethylation to normorphine and norcodeine, respectively. Normorphine is only 25% as active as analgesic as morphine [111]. According to DeRuiter [112], the decrease in the analgesic activity of normorphine compared to morphine is due to increased polarity with the consequent reduction in blood-brain barrier crossing. The same reasoning may be extrapolated to norcodeine reduced pharmacologic activity in comparison to codeine. The main route of morphine metabolism is glucuronidation at positions 3 and 6 (Figure 25) [113,114,115,116], with morphine-6-glucuronide accounting for the major part of analgesic activity of morphine [112]. In addition to *O*-demethylation to morphine, codeine is metabolized in a similar manner to morphine (Figure 25), with the 6-glucuronide conjugate playing the major role in analgesia and norcodeine playing only a little role [117]. The pKa and log *p* values of codeine and morphine and their *N*-desmethyl metabolites are given in Figure 25.

### 5.2. Tramadol

Tramadol (Figure 26) is a centrally acting analgesic that exerts its effect through two mechanisms: (a) as neurotransmitter reuptake inhibitor, and (b) as mu-receptor agonist [118]. It is of interest to note that tramadol has been designed as a congener of the mu-receptor agonist codeine [119]. Tramadol contains an aromatic methoxy group and an aliphatic dimethylaminomethyl group. The major pathways of tramadol metabolism are depicted in Figure 26 [120,121,122]. *O*-desmethyltramadol is the major active metabolite of tramadol; it acts mainly as mu-receptor agonist [122]. On the other hand, the *N*-desmethyl metabolite (nortramadol) is devoid of analgesic activity [122]. The pKa and log *p* values of tramadol and nortramadol are given in Figure 26. 

### 5.3. Propoxyphene

Propoxyphene (Figure 27) is a chiral drug whose dextro enantiomer (dextropropoxyphene) is an opioid mu-receptor agonist used as an analgesic in the treatment of mild to moderate pain [123]. It contains a dimethylaminopropyl moiety (Figure 27). The major route of metabolism of dextropropoxyphene is *N*-demethylation to norpropoxyphene (Figure 27) [123,124], which is active as a mu-receptor agonist and is used clinically. Dextropropoxyphene has been chosen as an example where metabolic *N*-dealkylation has led to the formation of a cardiotoxic metabolite with the consequent withdrawal of the drug in the US and Europe with restricted use in other countries [125]. It is interesting to note that despite the presence of an ester group and two unhindered phenyl groups in dextropropoxyphene, ester hydrolysis and aromatic-ring hydroxylation have not been reported as metabolic pathways of this drug. The pKa and log *p* values of propoxyphene and norpropoxyphene are given in Figure 27.

### 5.4. Meperidine (Pethidine)

Meperidine (Figure 28) is an opioid mu-receptor agonist used as analgesic. It contains a heterocyclic tertiary methylamino moiety (Figure 25). The metabolic pathways of meperidine are given in Figure 28 [126,127]. Normeperidine is as half as potent as meperidine but twice as active as CNS stimulant [106]. Further, neurotoxicity has led to restricted use of normeperidine as an opioid analgesic [128]. It is noteworthy that meperidine and normeperidine are also metabolized by ester hydrolysis to the corresponding inactive meperidine acids [127]. The pKa and log *p* values of meperidine and normeperidine are given in Figure 28.

## 6. Calcium Channel Blockers

### 6.1. Verapamil

Verapamil (Figure 29) is a phenylalkylamine calcium-channel blocker used in the treatment of hypertension, angina and cardiac arrhythmias [129]. It contains an internal tertiary methylamino moiety (Figure 29). The metabolic pathways of verapamil are depicted in Figure 29 [129,130,131,132,133]. Norverapamil retains 20% of cardiovascular activity of the parent drug [134]. On the other hand, *N*-desalkyl verapamil (D617) is presumably inactive due to the loss of a substantial (pharmacophoric) part of the molecule in the parent drug. The pKa and log *p* values of verapamil and norverapamil are given in Figure 29.

### 6.2. Diltiazem

Diltiazem (Figure 30) is a benzothiazepine calcium-channel blocker with antihypertensive and antiarrhythmic properties. It contains a dimethylaminoethyl moiety (Figure 30) and is primarily metabolized via the pathways shown in Figure 30 [134,135,136]. The four metabolites retain calcium channel blocking activity to varying extents, though less than the parent drug [135]. The pKa and log *p* values of diltiazem and *N*-desmethyldiltiazem are given in Figure 30.

### 6.3. Amiodarone

Amiodarone (Figure 31) is an antiarrhythmic drug used in the treatment of irregular heartbeats. It acts as a calcium-, potassium-, and sodium-channel blocker [137,138]. It contains a diethylaminoethyl moiety and is metabolized as shown in Figure 31 by CYP3A4 and CYP2C8 *N*-demethylation to *N*-desethylamiodarone, which is active as antiarrhythmic [139,140,141]. The pKa and log *p* values of amiodarone and *N*-desethylamiodarone are given in Figure 31.

## 7. Drugs Acting on Sodium Channels

### 7.1. Local Anesthetics

Local anesthetics produce anesthesia by inhibiting excitation of nerve endings or by blocking conduction in peripheral nerves [142]. Prilocaine binds to the intracellular surface of sodium channels, which blocks the subsequent influx of sodium into the cell. Action potential propagation and nerve function is, therefore, prevented. This block is reversible and when the drug diffuses away from the cell, sodium channel function is restored and nerve propagation returns [143].

#### 7.1.1. Lidocaine

Lidocaine (Figure 32) is a local anesthetic antiarrhythmic drug. As a local anesthetic, it belongs to the amino-amide class. Due to the steric protective effect provided by the two *ortho*-methyl groups on the benzene ring, possible amide hydrolysis of lidocaine is excluded leaving only *N*-deethylation as the viable major metabolic route. The metabolic products of lidocaine are monoethylglycinexylidide (MEGX), which is active as local anesthetic with a longer duration of action than lidocaine, and glycinexylidide and lidocaine-*N*-oxide, which are inactive (Figure 32) [144,145,146,147]. The pKa and log *p* values of lidocaine and monoethylglycinexylide are given in Figure 32.

## 8. Drugs That Act on GABA_A_nergic Receptor

### 8.1. Zopiclone

Zopiclone (Figure 33) is a cyclopyrrolone derivative with hypnotic effects. Its mechanism of action involves increase in the normal transmission of GABA in the CNS via modulating benzodiazepine receptors in the same way that benzodiazepine drugs do [148]. Zopiclone contains a heterocyclic tertiary methylamino moiety (Figure 33). It is metabolized as per the pathways depicted in Figure 33 to give *N*-desmethylzopiclone, which is as active as hypnotic as zopiclone and zopiclone-*N*-oxide, which is devoid of hypnotic activity [149,150,151,152,153]. The pKa and log *p* values of zopiclone and *N*-desmethylzopiclone are given in Figure 33.

## 9. Muscarinic Receptor Blockers

### 9.1. Tolterodine/Fesoterodine

Tolterodine (Figure 34) is an antimuscarinic drug used in the treatment of overactive bladder (OAB) and urinary urge incontinence. It contains a diisopropylaminopropyl moiety. As shown in Figure 34, tolterodine is metabolized through monodeisopropylation to give an inactive metabolite and through benzylic-methyl group oxidation to give 5-hydroxymethyl tolterodine (5-HMT), which is equiactive with the parent drug [154,155,156,157]. Despite being equiactive to its parent drug, 5-HMT did not qualify to the status of metabolite drug because of its high hydrophilicity (log *p* value of 0.73) and, accordingly, poor bioavailability [155]. However, the problem has been resolved by esterifying the aromatic hydroxy (phenolic) group with isobutanoic acid to give the prodrug fesoterodine of log D_7.4_ of 5.7 [156] with the consequent substantial improvement of bioavailability. Fesoterodine is also metabolized by CYP3A4 to *N*-desisopropylfesoterodine [158], which is presumably inactive in analogy with desisopropyltolterodine.

The pKa and log *p* values of tolterodine and fesoterodine together with the log *p* value of 5-hydroxymethyltolterodine are given in Figure 34. No corresponding pKa and log *p* values have been found for *N*-desisopropyltolterodine or *N*-desisopropylfesoterodine; however, they can be estimated to be respectively higher and lower than those of tolterodine.

### 9.2. Oxybutynin

Oxybutynin (Figure 35) is a chiral antimuscarinic drug used in the treatment of overactive bladder. The *R*-enantiomer of oxybutynin accounts for all the antimuscarinic activity while the *S*-enantiomer is inactive [159]. The drug contains a diethylaminobutynyl moiety and is metabolized through the pathways depicted in Figure 35 [160,161,162,163,164]. *N*-desethyloxybutynin has similar activity to oxybutynin as antimuscarinic [159]. The pKa and log *p* values of oxybutynin and *N*-desethyloxybutynin are given in Figure 35.

## 10. “If” Channel Blockers

### 10.1. Ivabradine

Ivabradine (Figure 36) is used for the symptomatic management of stable heart-related chest pain and heart failure not fully managed by beta-blockers. Ivabradine lowers heart rate by selectively inhibiting If channels (“funny channels”) in the heart in a concentration-dependent manner without affecting any other cardiac ionic channels (including calcium or potassium) [165]. The drug contains a tertiary methylamino moiety and is metabolized predominantly in the liver and intestines by the cytochrome P450 (CYP) 3A4 enzyme to active *N*-desmethylivabradine (S-18982), which circulates at concentrations of approximately 40% [166]. The pKa and log *p* values of ivabradine are given in Figure 36. No corresponding data have been found for *N*-desmethylivabradine; however, they can be predicted as, respectively, higher and lower than those of ivabradine.

## 11. Drugs That Act as Enzyme Inhibitors

### 11.1. Sildenafil

Sildenafil (Figure 37) is a member of a class of medications called phosphodiesterase (PDE) inhibitors. It is used to treat erectile dysfunction in men as well as pulmonary arterial hypertension [167]. It contains a heterocyclic tertiary methylamino moiety, which represents the site of metabolism upon *N*-demethylation to give *N*-desmethylsildenafil (Figure 37) [168,169,170,171]. The metabolite possesses a PDE5 selectivity that is similar to the parent sildenafil molecule and in vitro potency for PDE approximately 50% that of the parent drug; it accounts for 20% pharmacologic activity of sildenafil [171]. The pKa and log *p* values of sildenafil and *N*-desmethylsildenafil are given in Figure 37.

## 12. Drugs That Act on Microorganisms

### 12.1. Chloroquine/Hydroxychloroquine

Chloroquine (CQ) and hydroxychloroquine HCQ) (Figure 38 and Figure 39, respectively) are aminoquinolones that inhibit polymerase; they are used in the treatment and prophylaxis of malaria. In order to stop malaria, they cause the accumulation of heme, which is toxic and deadly to the parasite. The heme is accumulated due to the inhibition of heme polymerase that takes place [172]. However, the use of CQ and HCQ in the treatment and prophylaxis of malaria has declined because of development of resistance [173]. Currently, both CQ and HCQ have made a notable comeback in chemotherapy in the treatment of Covid-19. Drugs repurposing (adaptation for use in a different purpose) to fall into the treatment regime of COVID-19, are currently being tested. They fall into one of the two following categories: (i) drugs that target the replication cycle of the virus, and (ii) drugs that aim at controlling the disease’s symptoms. In view of the treatment for COVID-19, it is suggested that the CQ and HCQ work by inhibiting the entry of the virus into the host cells. The mechanism involves blocking the host receptors’ glycosylation, along with breaking down the formation of the virus proteins by inhibiting endosomal acidification by virtue of the drugs being basic in character, since each drug contains two basic nitrogens (i.e., each drug is a diacidic base) [174,175,176,177,178].

Even though there is a clear lack of adequate evidence of benefit of the drugs, many African and other countries have endorsed hydroxychloroquine repurposed (off-label) use for the treatment of COVID-19 contrary to the WHO recommendations [179]. On the other hand, the US Food and Drug Administration have also issued an Emergency Use Authorization for the use of chloroquine and hydroxychloroquine for the treatment of Covid-19 in adult populations [180].

Chloroquine contains a diethylaminopentyl moiety and is metabolized by sequential *N*-deethylation to *N*-desethylchloroquine and *N*,*N*-didesethylchloroquine as depicted in Figure 38 [181,182,183,184,185,186,187]. The two metabolites are respectively formed in 40% and 10% yields with respect to chloroquine. Both chloroquine and desethylchloroquine concentrations decline slowly, with elimination half-lives of 20 to 60 days. Both parent drug and metabolite can be detected in urine months after a single dose [164]. Interestingly, one literature report [177] mentions HCQ as a metabolite of CQ; however, this statement has not been substantiated by other reports on CQ metabolism. On the other hand, hydroxychloroquine contains ethyl/hydroxyethylene groups and is metabolized by sequential removal of the two groups as shown in Figure 39 [174,188,189].

The pKa and log *p* values for chloroquine and hydroxychloroquine are given in Figure 38 and Figure 39, respectively. Despite lack of pKa and log *p* values for desethylhydroxy chloroquine in the literature, predictions can be made. Secondary amino groups are invariably more basic and more polar with higher pKa and lower log *p* values than tertiary amino groups. Extrapolation can be extended to hydroxychloroquine and desethylhydroxychloroquine.

## 13. Anticancer Drugs

### 13.1. Imatinib

Imatinib (Figure 40) is first-line therapy for the treatment for all phases of chronic myelogenous leukemia and metastatic and unresectable malignant gastrointestinal stromal tumors [190]. Imatinib contains a heterocyclic tertiary methylamino moiety, which is metabolized by CYP3A4 oxidative *N*-demethylation to give *N*-desmethylimatinib, which is of similar potency to the parent drug [191,192,193]. According to Foye (2013) [192], the *N*-methyl substituent on the piperazine ring in imatinib has the role of increasing the water solubility and bioavailability profile of the drug; i.e., it plays an auxophoric role. The pKa and log *p* values of imatinib and *N*-desmethylimatinib are given in Figure 40.

### 13.2. Dacarbazine

Dacarbazine (Figure 41) is an anticancer alkylating prodrug used in the treatment of Hodgkin’s lymphoma, metastatic melanoma and soft tissue sarcoma [194,195]. Generation of the alkylating species, methyl diazonium, from dacarbazine occurs through a combination of metabolic processes including *N*-demethylation as a first step followed by tautomerization and spontaneous cleavage as shown in Figure 38 [195].

### 13.3. Tamoxifen

Tamoxifen (Figure 42) is an antiestrogen, which acts as anti-breast cancer by competitively blocking the estrogen receptor [196]. It contains a dimethylaminoethoxy moiety and is metabolized through the pathways depicted in Figure 42 [197,198,199,200,201,202,203,204]. While the binding affinity of 4-hydroxytamoxifen to the estrogenic receptor is 30–100 fold stronger than that of tamoxifen, the *N*-desmethyl metabolite binding affinity is less than that of tamoxifen; the *N*,*N*-didesmethyl metabolite has even less binding affinity than the *N*-desmethyl metabolite [199]. Further, containing a tertiary dimethylamino moiety, tamoxifen is metabolized by *N*-oxidation to tamoxifen-*N*-oxide, which is devoid of estrogen-receptor blocking activity. The pKa and log *p* values of tamoxifen, *N*-desmethyltamoxifen, 4-hydroxytamoxifen (afimoxifene) and 4-hydroxy-*N*-desmethyltamoxifen (endoxifen) are given in Figure 42.

### 13.4. Tormifene

Tormifene (Figure 43) is a first-generation nonsteroidal selective estrogen receptor modulator [205]. It has beneficial effects on the bone, and cardiovascular system; besides, it increases HDL levels [205]. Its structure is very similar to that of tamoxifen; the two drugs differ only in a chloro group in the side ethyl chain of tormifene. Similar to tamoxifen, tormifene contains a dimethylaminoethoxy chain where metabolic changes occur as depicted in Figure 40. Analogous to tamoxifen, tormifene is metabolized to *N*-desmethyltormifene, 4-hydroxytormifene-*N*-desmethyltormifene tormifene-*N*-oxide [206,207,208] and ospermifene [209] as depicted in Figure 43. Although the activities of the first two metabolites relative to the tormifene are not reported in the literature, they can be inferred from the activities of metabolites of the closely related drug tamoxifen: *N*-desmethyltormifene is expected to have little activity while 4-hydroxy-*N*-desmethyltormifene has significant activity. Ospermifene is a selective estrogen-receptor modulator [205]. The pKa and log *p* values of tormifene are given in Figure 43. The corresponding values for the *N*-desmethyl metabolite of tormifene were not available, but can be inferred as higher as and lower than those of tormifene, respectively.

## 14. Metabolic *N*-Dealkylation and *N*-Oxidation

### 14.1. Metabolic N-Dealkylation

The following observations can be made from the cited drug cases:(1)In drugs containing aliphatic open-chain tertiary *N*,*N*-dialkylamino moieties, the alkyl groups are methyl, ethyl or isopropyl.(2)In drugs containing heterocyclic tertiary *N*-alkylamino moieties, the ring is either piperidine or piperazine and the alkyl group is invariably methyl.(3)*N*-dealkylation of aliphatic tertiary *N*,*N*-dialkylamino moieties is sequential for some drugs giving rise to secondary *N*-alkylamino moieties and primary amino groups.(4)As far as pKa values are concerned, *N*-dealkylation of *N*-alkylamino moieties invariably results in situations where pKa (3° amine) < pKa (2° amine) > pKa (1° amine) for all the reviewed drug cases.(5)For log *p* values, the corresponding order is log *p* (3° amine) > log *p* (2° amine) > log *p* (1° amine).(6)The *N*-desalkyl metabolites of tertiary and secondary-alkylamino-moiety-containing parent drugs vary in pharmacologic activities being more active, equiactive (sometimes with alteration in the mechanism of action), less active or inactive.

The pKa and log *p* values of the parent drugs and their *N*-desalkyl metabolites have been obtained from DrugBank [210]. Where the corresponding pKa and log *p* data are not available for the *N*-desalkyl metabolites, they can generally be inferred as higher and lower, respectively, than those of the parent drugs.

The order of the pKa values of the three amine classes is pKa (*N*,*N*-dialkylamino) < pKa (*N*-monoalkylamino) > (primary amino), and is explicable by electronic and steric effects [211,212]. On the other hand, the order of the corresponding log *p* values is log *p* (*N*,*N*-dialkylamino) > log *p* (*N*-monoalkylamino) > log *p* (primary amino), which is due to reduced polarity and hence reduced water solubility and enhanced lipid solubility of the amino-group-containing compound from left to right.

The electronic and steric effects on *N*-alkylamino moieties can be explained as thus: alkyl groups (such as methyl, ethyl and isopropyl) are electron-donating (or electron-releasing) groups. Hence, in *N*-alkylamino moieties, the alkyl groups will tend to increase the electron density on the nitrogen, rendering it more basic (i.e., with higher pKa value). The increase of basicity will lead to increase in the concentration of the protonated (ionized) form of the compound relative to the unionized form as can be calculated by the Henderson-Hasselbalch equation [213]. On the other hand, however, when an alkyl group replaces the hydrogen atom of the secondary *N*-alkylamino moiety it will exert a steric effect. This steric effect will hinder the approach of a proton (H^+^) to access the lone pair of electrons on the nitrogen of the resulting tertiary *N*,*N*-dialkylamino moiety. The result of the steric effect is hence decrease of the basicity of *N*,*N*-dialkylamino moieties relative to *N*-monoalkylamino moieties. In summary, the electronic effect is manifest in secondary and tertiary alkylamino moieties relative to primary amino moieties, while the steric effect explains the decrease of the basicity of tertiary *N*,*N*-alkylamino moieties relative to secondary *N*-alkylamino moieties. Therefore, the implication of the electronic and steric effects is that the secondary alkylamino moieties in *N*-monoalkylamino metabolites will be more protonated (ionized) than tertiary *N*,*N*-alkylamino moieties in the parent drugs. Accordingly, if the ionic (salt bridge) binding of the alkylamino moiety to the receptor is an essential pharmacophoric character, then with equimolar amounts of the parent drug and its *N*-desalkyl metabolite, the latter should have more affinity to the receptor and possibly higher efficacy than its parent drug. As well, hydrogen-bonding interactions are more manifest in secondary *N*-alkylamino moieties, which act as both hydrogen-bond acceptors and donors, as compared to tertiary *N*,*N*-dialkylamino moieties, which only act as relatively weak hydrogen-bond acceptors. Accordingly, assuming that ionic and hydrogen-bond bindings play a crucial role in determining the affinity of alkylamino-moiety-containing drugs, one would expect the secondary alkylamino metabolites to be more active than the tertiary-alkylamino-moiety-containing parent drugs. However, in all the cited drug cases in this review this has not been observed to be the case as factors other than affinity to drug receptors govern the pharmacologic activity of metabolites as will be discussed in due course. Shein and Smith (1978) [214] stated that in TCAs, amine substitution by alkyl groups does not alter ionization of the nitrogen in both imipramine and desmipramine “as both compounds have pKa values of 9.5”. To quote the authors [214], “*Despite the high percentage ionization of this group (the monoalkylamino in desmipramine or dialkylamino in imipramine) at the pH of the Tyrode solution (presumably pH 7.4), attachment of the terminal part of the side chain is largely nonpolar in type*”. We tend to differ with this statement [214], which may not be, in totality, true as the steric effect in the tertiary dimethylamino moiety in imipramine entails lower basicity (lower pKa) than the secondary alkylamino moiety in desmipramine as is evident from the pKa values of the two drugs, 9.2 and 10.02, respectively. We argue that the amino groups play a significant role in the attachment of the side chain to the receptor through ionic and hydrogen-bonding interactions. This argument is substantiated by literature reports [215,216,217].

Further, the decrease of log *p* of secondary-alkylamino-moiety-containing metabolites relative to tertiary- alkylamino-moiety-containing parent drugs is due to the ability of the secondary alkylamino moieties to act as both hydrogen-bond donors and acceptors while the tertiary alkylamino moieties act as only hydrogen-bond acceptors. Accordingly, the secondary alkylamino moieties are able to establish more hydrogen bonds with water than the tertiary alkylamino moieties. The inference is that the aqueous solubility of the secondary alkylamino metabolites will increase with the subsequent decrease of log *p* relative to the tertiary alkylamino parent drugs. The literature log *p* values of the dialkylamino parent drugs and the alkylamino metabolites given in the figures are invariably in line with the above prediction.

Both the effects of pKa and log *p* modifications by metabolic *N*-dealkylation tend to impede penetration of the *N*-desalkylamino metabolites of lipophilic cell membranes, thus leading to decrease in the effective concentrations of the metabolites at the receptor resulting in attenuation of pharmacologic activity in most of the cited drug cases. The attenuation of the pharmacologic activity of the *N*-desalkylamino metabolites may occur despite the fact that they may have stronger affinities for the receptor than the dialkylamino-moiety-containing parent drugs as has been argued earlier. Sahu et al. [218] have associated the decrease of log *p* of the anti-HIV tetrahydroimidazobenzodiazepinones with decrease in pharmacologic activity, however, without explicitly giving the reason.

#### 14.1.1. Focused N-Dealkylation Cases

Attenuation or retaining of pharmacologic activity has been observed for most of the *N*-monodesalkyl metabolites of the drug cases cited in this review and are explicable by the physicochemical differences between the metabolites and parent drugs. However, other cases, which help in elucidating the role of alkylamino moieties in drug molecules acting at various receptors have also been observed and are thus focused.

##### 14.1.1.1. Loss of Pharmacologic Activity

Loss of pharmacologic activity of the *N*-monodesalkyl metabolites with respect to the *N*,*N*-dialkylamino parent drugs has been reported for some cases. These drugs include the antidepressant venlafaxine (Section 2.1.3, Figure 6) [219,220], the analgesic tramadol (Section 5.2, Figure 23) [221] and the antimuscarinic tolterodine (Section 9.1, Figure 31) [222]. Loss of pharmacologic activity is usually associated with loss or modification of a pharmacophoric structural feature in the original drug molecule. We therefore argue that since *N*-alkyl groups in drug molecules do not form hydrogen or ionic bonds with receptors, abolishment of pharmacologic activity upon their loss by metabolic *N*-dealkylation is explicable by both alkyl groups in *N*,*N*-dialkylamino moieties in venlafaxine, tramadol and tolterodine playing primary pharmacophoric roles. The binding of the two-alkyl groups to the receptors via van der Waals forces is crucial for affinity and accordingly to the efficacy and activity of the parent drugs. The same phenomenon may be extrapolated to secondary alkylamino parent drugs upon their metabolic *N*-dealkylation to primary amines. Further, the alkyl groups of the *N.N*-dialkylamino moieties in venlafaxine, tramadol and tolterodine may play a logistic pharmacophoric role of orienting the protonated nitrogen of the alkylamino moieties for optimum binding to the aspartate amino-acid residues in the corresponding receptors [223,224].

On the other hand, three drugs have been noted for the complete metabolic loss of the pharmacophoric alkylamino moieties with the consequent loss of pharmacologic activity: diphenhydramine to diphenylmethoxyacetic acid (Section 4.1, Figure 22), oxybutynin to 2-cyclohexyl-2-phenylglycolic acid (Section 9.2, Figure 35) and tormifene to ospermifene (Section 13.4, Figure 43). In tormifene, the metabolic loss of the alkylamino moiety has led to alteration of the mechanism of the metabolite, ospermifene, to modulator rather than blocker of the estrogenic receptor as is the case with the parent drug. In addition, diphenhydramine is metabolized to inactive *N*-acetyl-*N*-desmethyldiphenhydramine in which the amide group is only capable of hydrogen-bond binding to the receptor but not of ionic binding since it (the amide group) is not ionizable. The latter observation gives supporting evidence to the importance of receptor ionic binding of the *N*,*N*-dimethylamino and *N*-methylamino moieties in the parent drug and metabolite, respectively.

##### 14.1.1.2. Modification of Receptor Inhibition Selectivity

Metabolic *N*-demethylation of tertiary dimethylamino moieties has resulted in the modification of receptor inhibition selectivity as exemplified by the TCAs imipramine to desmipramine and amitriptyline to nortriptyline (Section 2.1.1, Figure 3 and Figure 4, respectively). The parent drugs, imipramine and amitriptyline, are more selective inhibitors of serotonin transport receptor (SET) than norepinephrine transport receptor (NET) while the opposite effect is true for the respective metabolite drugs, desmipramine and nortriptyline. This shift in receptor inhibition selectivity may be explained by two possibilities. Firstly, the two pharmacophoric methyl groups in the parent drugs (imipramine and amitriptyline) bind to the SET receptor via van der Waal’s forces as opposed to the one-methyl-group binding in the *N*-desmethyl metabolite drugs (desmipramine and nortriptyline). Secondly, the hydrogen bond and ionic bindings are more manifested in the *N*-desmethyl metabolite drugs to the NET receptor relevant to the parent drugs. According to Goral et al. [223], the methyl groups in the dimethylamino moiety may help in orienting the protonated amino groups in the drug and metabolites for optimum ion-ion binding with the receptor. A quote from Goral et al.’s paper [223] is thus, “*Aspartic acid D75 plays a key role in recognition of the basic amino group present in monoamine transporter inhibitors and substrates*”. Substantiating evidence in this respect is found in the work by Patil et al. [224] and Maria et al. [225]. A quote from Patil et al.’s paper [224] is thus: “*The results presented here demonstrate that hydrogen bonding and optimized hydrophobic interactions both stabilize the ligands at the target site, and help alter binding affinity and drug efficacy*”. A quote from López-Rodríguez et al.’s paper [225] is thus, “*Serotonin transporter receptor ligands docking: Forty-five structurally diverse 5-hydroxytryptamine6 receptor (5-HT6R) antagonists were selected to develop a 3D pharmacophore model with the Catalyst software. The structural features for antagonism at this receptor are a positive ionizable atom interacting with Asp3.32, a hydrogen bond acceptor group interacting with Ser5.43 and Asn6.55, a hydrophobic site interacting with residues in a hydrophobic pocket between transmembranes 3, 4, and 5, and an aromatic-ring hydrophobic site interacting with Phe6.52*”.

##### 14.1.1.3. Activation of Prodrugs

An example where metabolic *N*-demethylation of *N*,*N*-dimethylamino moiety has resulted in the active form of the drug is dacarbazine (an anticancer drug), which is transformed to the DNA-alkylating entity methyl diazonium. Methyl diazonium results from the successive processes of metabolic *N*-demethylation, tautomerization and spontaneous cleavage of the prodrug dacarbazine as shown in Figure 41. Thus, dacarbazine is a prodrug that is activated in vivo by metabolic and chemical processes. The first crucial step is the metabolic *N*-demethylation.

##### 14.1.1.4. Potential Drug Candidates (Metabolite Drugs)

Metabolic *N*-demethylation as well as 4-hydroxylation are essential steps in the formation of active forms of the breast cancer drug, tamoxifen (Section 13.3, Figure 42). 4-Hydroxytamoxifen (afimoxifene) [226] and *N*-desmethyl-4-hydroxytamoxifen (endoxifen) [200] are presently in the final phases of development as drugs for the treatment of breast cancer. After they have qualified for clinical use, the two candidate drugs will bypass the use of the prodrug tamoxifen and will be of benefit for breast-cancer patients who lack the enzyme CYP2D6, which activates tamoxifen in vivo.

Chloroquine and hydroxychloroquine are used as antivirals in some drug treatment protocols of Covid-19. The mechanism of the antiviral action of both drugs has been suggested as being due to the inhibition of endosomal acidification by virtue of the basicity of the two drugs since each of them contains two basic nitrogens, i.e., the two drugs are diacidic bases (Section 12.1, Figure 38 and Figure 39) [175,176,177,178,179]. If this theory of the mechanism of action were to be endorsed, then the *N*-desethyl metabolites of the two drugs are expected to be more efficacious as antiviral agents than the parent drugs due to their higher basicity. The fact that 40% of a dose of chloroquine is metabolized to *N*-desethylchloroquine consolidates the possible participation of the *N*-desethyl metabolite in the antiviral activity based on the basicity theory. Accordingly, thought may have to be given to consider the *N*-desethyl metabolites of chloroquine and hydroxychloroquine as potential drug candidates and develop them into fully-fledged drugs against Covid-19.

### 14.2. N-Oxidation of Tertiary-Alkylamino-Moiety-Containing Drugs

The following observations have been made from the cited drug cases metabolized by *N*-oxidation:(1)*N*-oxidation has been observed for only tertiary alkylamino groups (aliphatic open-chain or heterocyclic), but not for secondary alkylamino or primary amino groups.(2)Regarding pharmacologic activity, *N*-oxides of all the reviewed drug cases are inactive.(3)Metabolically formed *N*-oxides of drugs may revert to the parent drugs through bioreduction.

The structure of the *N*-oxide group and its representations are shown in Figure 44.

*N*-oxidation is a common metabolic pathway of most drugs containing aliphatic and heterocylic tertiary alkylamino moieties [227]. All the *N*-oxide metabolites of the reviewed drug examplesare devoid of pharmacologic activity. The lack of pharmacologic activity of the *N*-oxides is a result of the masking of the potential cationic charge of the amine, which abolishes its potential ability to interact with receptors through ionic bindings. In case the ionic binding of the amine is primary pharmacophoric, its loss should explains why *N*-oxides of drugs are inactive. Nevertheless, upon bioreduction of the *N*-oxide metabolite to the tertiary-amino-moiety-containing drug, the amino group will resume its ability to be protonated (i.e., be ionized) at pH 7.4 and establish ionic binding essential for drug-receptor affinity and accordingly drug efficacy and activity.

By being converted back to the active forms, the *N*-oxide metabolites of drugs have been suggested as bioreductive prodrugs [228,229,230]. The planning of *N*-oxides as prodrugs implies that the *N*-oxides are devoid of pharmacologic activity and need to be bioactivated by Flavin-containing monooxygenase (FMO) in vivo. In fact, some *N*-oxide prodrugs are currently marketed while others have been patented. The marketed *N*-oxide prodrugs are imipraminoxide (the *N*-oxide of imipramine [231], Figure 3) and amitriptylinoxide (the *N*-oxide of amitriptyline, Figure 4) [232]. They are used for the treatment of depression; they have similar effects as well as equivalent efficacy to their active forms. The patented *N*-oxide prodrugs include:
(i)Sildenafil-*N*-oxide, a prodrug of sildenafil, described in a patent for the treatment of erectile dysfunction and pulmonary arterial hypertension (PAH) [233].(ii)Venlafaxine-*N*-oxide and *O*-desmethylvenlafaxine-*N*-oxide, both of which have been patented as prodrugs of venlafaxine and *O*-desmethylvenlafaxine, respectively, and are used in the treatment of depression [234].(iii)Lidocaine (lignocaine) *N*-oxide used in the treatment of pulmonary inflammation associated with asthma, bronchitis, and chronic obstructive pulmonary disease (COPD) [235].


The *N*-oxide metabolites of the tertiary alkylamino drugs cited in this review form potential candidates for prodrug development with possible improved bioavailability and longer duration of action relative to the parent drugs. The *N*-oxide metabolites include clomipramine-*N*-oxide, doxepin-*N*-oxide, citalopram-*N*-oxide, clozapine-*N*-oxide, mirtazapine-*N*-oxide and olanzapine-*N*-oxide.

## 15. Conclusions

Alkylamino moieties in drug molecules undergo two types of metabolic reactions: *N*-dealkylation and *N*-oxidation. The former metabolic change has resulted in clinically used drugs, potential drugs, activation of prodrugs as well as attenuation and loss of activity of drugs. The *N*-oxide metabolites resulting from *N*-oxidation of dialkylamino moieties are invariably pharmacologically inactive but are bioreducible to the active forms. As thus, they have formed and will form basis of prodrug development. The physicochemical changes that result from *N*-dealkylation and *N*-oxidation of alkylamino moieties explain the changes in the metabolites relative to the parent drugs regarding binding to receptors, affinity, efficacy and accordingly pharmacological activity. The information provided is of broad utility in structure-based drug design.

## Figures and Tables

**Figure 1 molecules-26-01917-f001:**
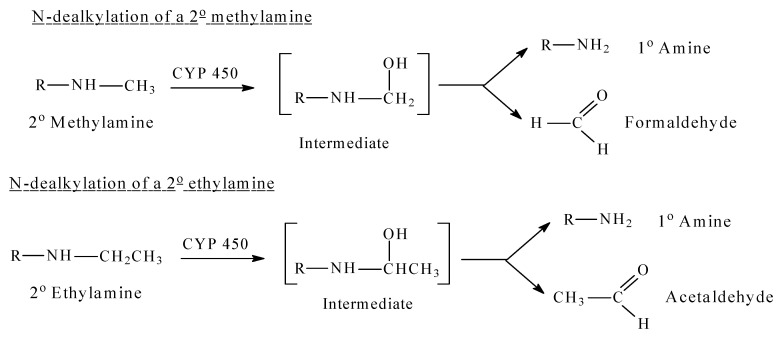
CYP 450 oxidative dealkylation of alkylamines.

**Figure 2 molecules-26-01917-f002:**
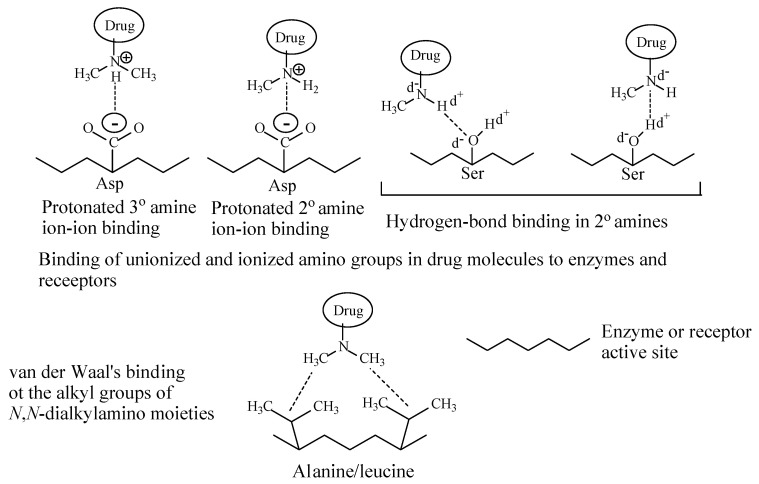
Alkylamino moiety binding to receptors.

**Figure 3 molecules-26-01917-f003:**
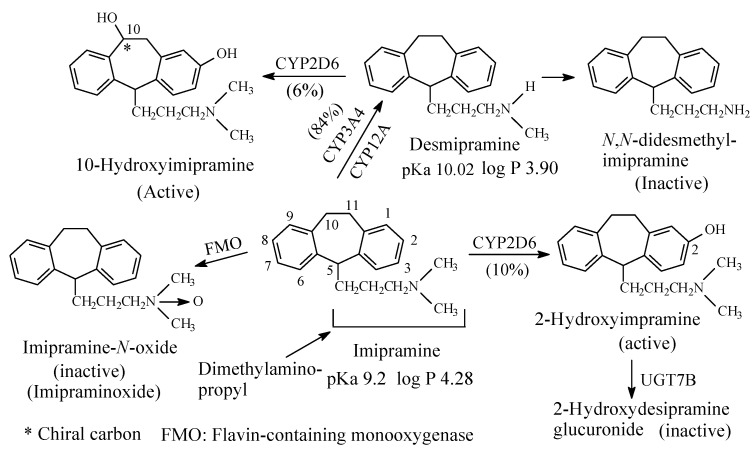
Metabolic pathways of imipramine.

**Figure 4 molecules-26-01917-f004:**
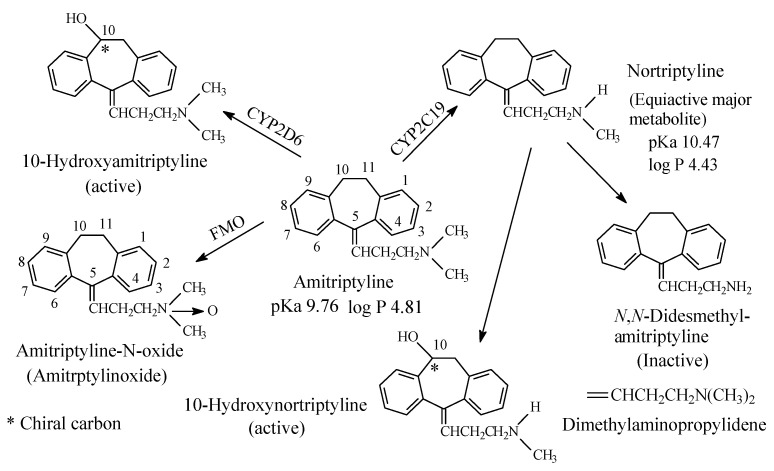
Metabolic pathways of amitriptyline.

**Figure 5 molecules-26-01917-f005:**
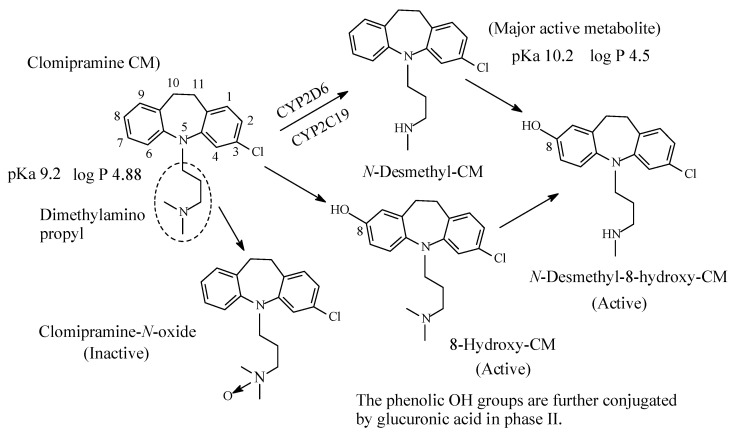
Metabolic pathways of clomipramine.

**Figure 6 molecules-26-01917-f006:**
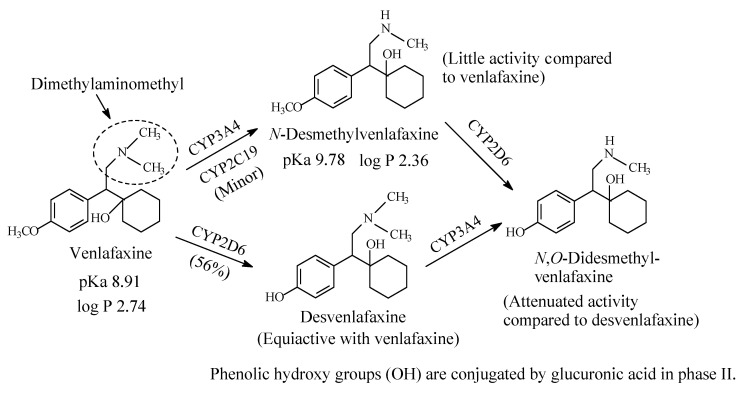
Major metabolic pathways of venlafaxine.

**Figure 7 molecules-26-01917-f007:**
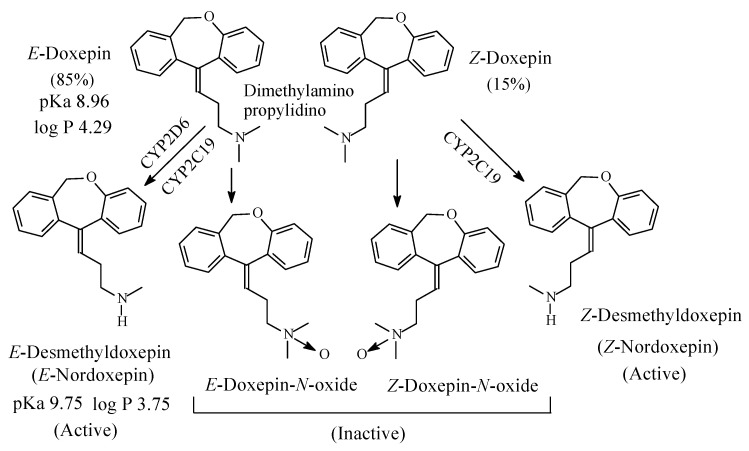
Metabolic pathways of doxepin.

**Figure 8 molecules-26-01917-f008:**
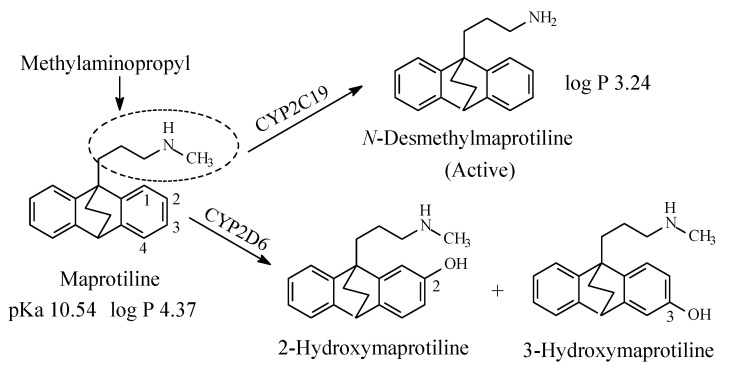
Major metabolic pathways of maprotiline.

**Figure 9 molecules-26-01917-f009:**
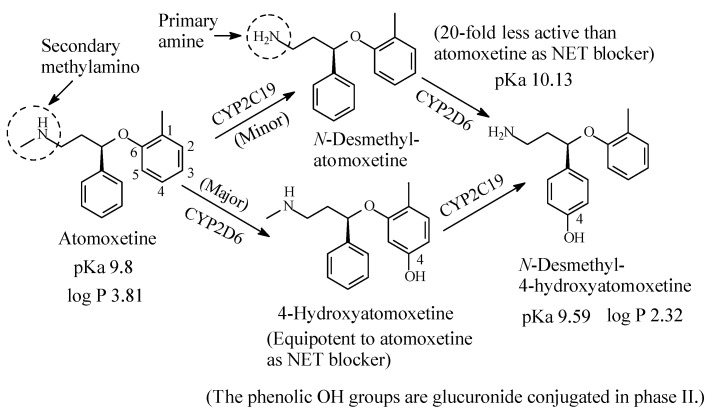
Metabolic pathways of atomoxetine.

**Figure 10 molecules-26-01917-f010:**
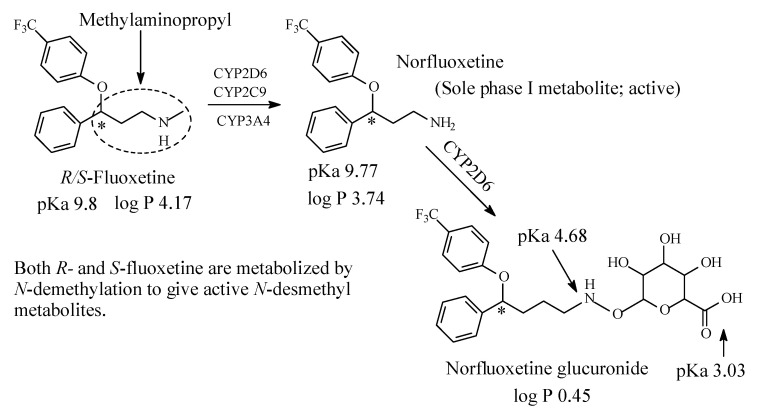
Metabolic pathway of fluoxetine.

**Figure 11 molecules-26-01917-f011:**
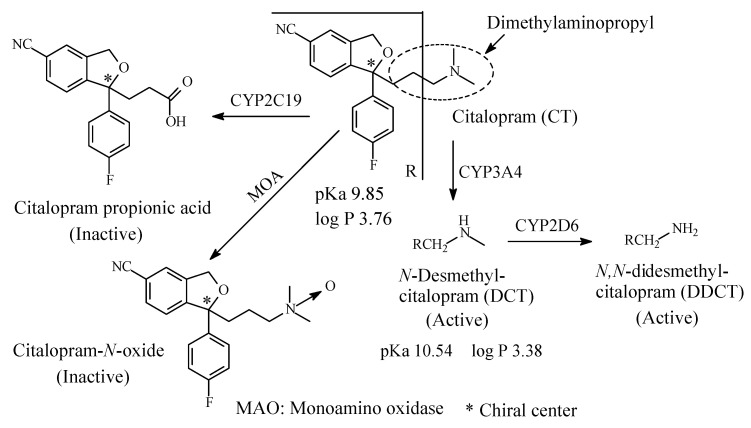
Metabolic pathways of citalopram.

**Figure 12 molecules-26-01917-f012:**
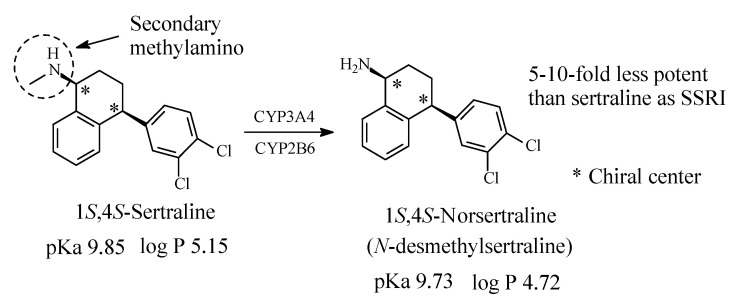
Metabolism of sertraline.

**Figure 13 molecules-26-01917-f013:**
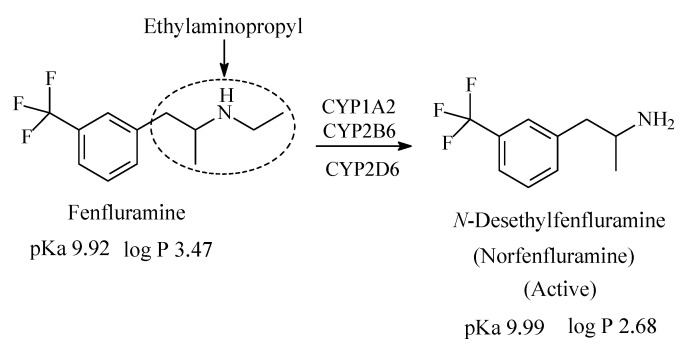
Metabolism of fenfluramine.

**Figure 14 molecules-26-01917-f014:**
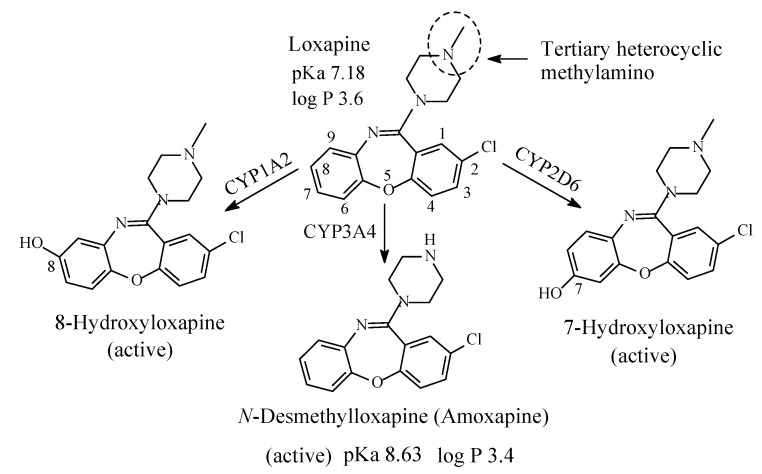
Metabolic pathways of loxapine.

**Figure 15 molecules-26-01917-f015:**
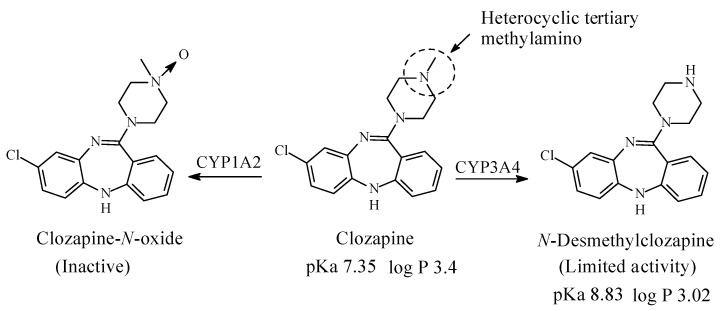
Metabolic pathways of clozapine.

**Figure 16 molecules-26-01917-f016:**
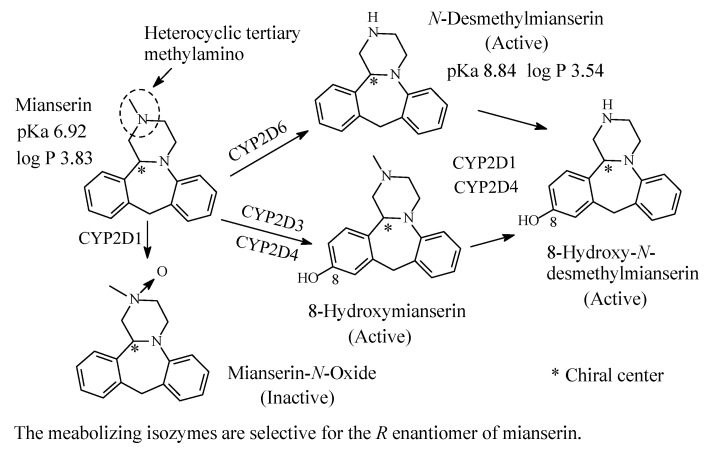
Metabolic pathways of mianserin.

**Figure 17 molecules-26-01917-f017:**
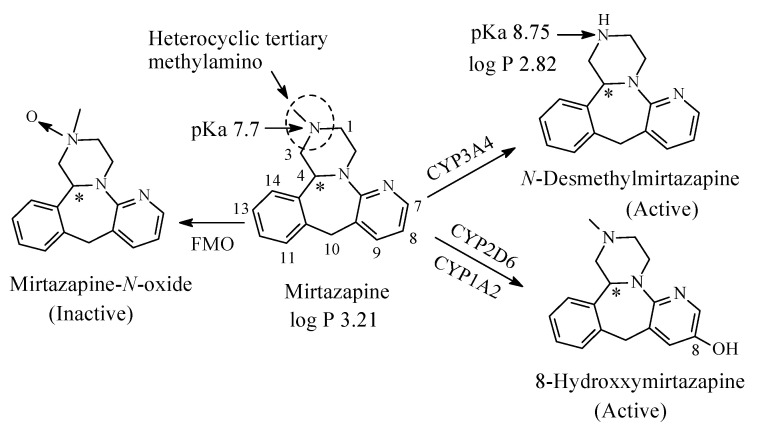
Metabolic pathways of mirtazapine.

**Figure 18 molecules-26-01917-f018:**
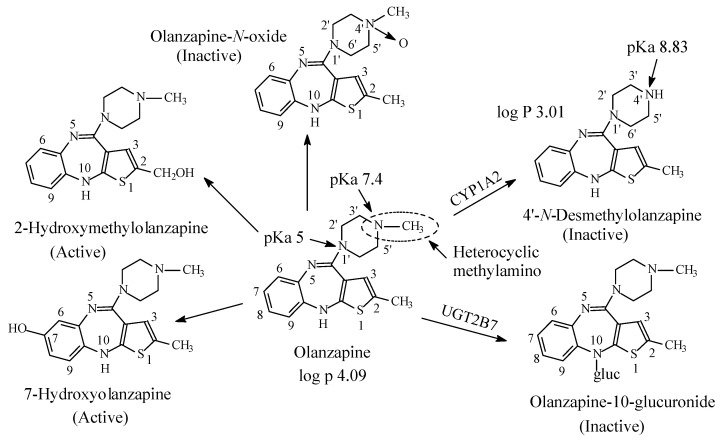
Metabolic pathways of olanzapine.

**Figure 19 molecules-26-01917-f019:**
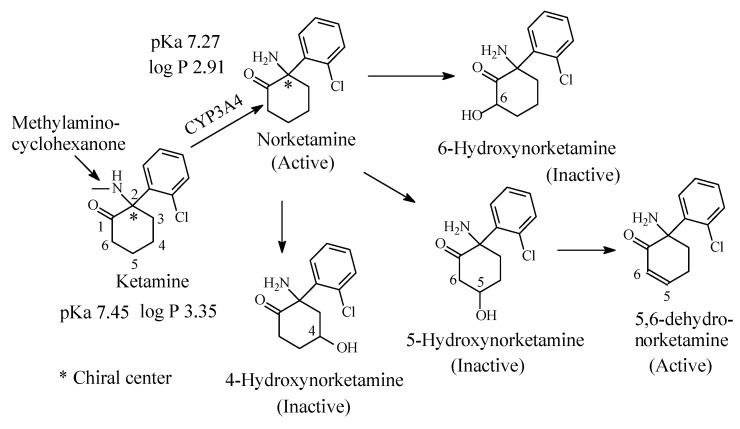
Metabolic pathways of ketamine.

**Figure 20 molecules-26-01917-f020:**
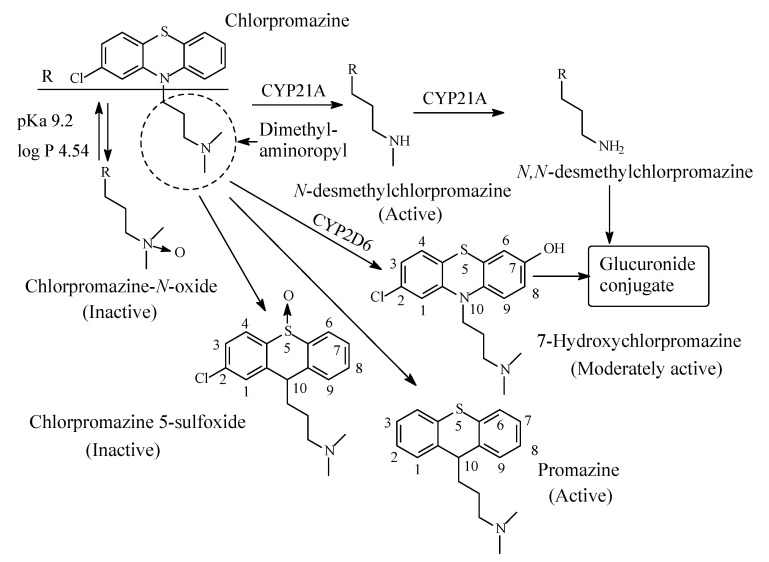
Major metabolic pathways of chlorpromazine.

**Figure 21 molecules-26-01917-f021:**
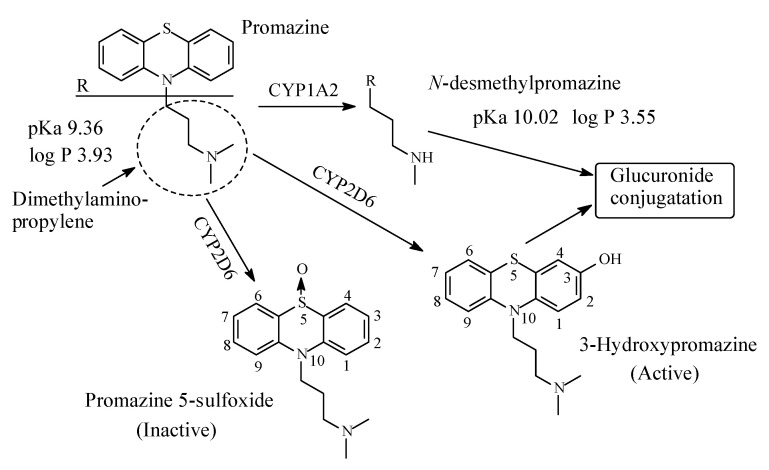
Metabolic pathways of promazine.

**Figure 22 molecules-26-01917-f022:**
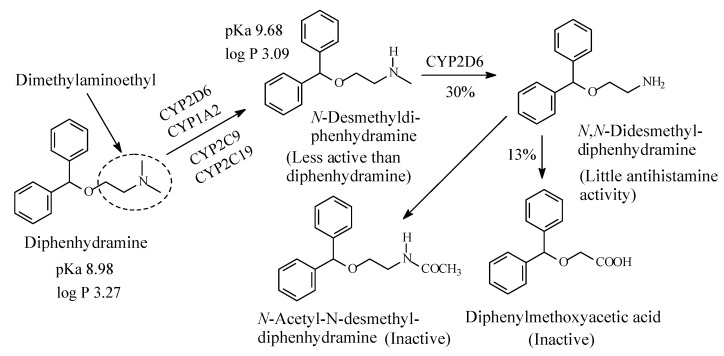
Metabolic pathways of diphenhydramine.

**Figure 23 molecules-26-01917-f023:**
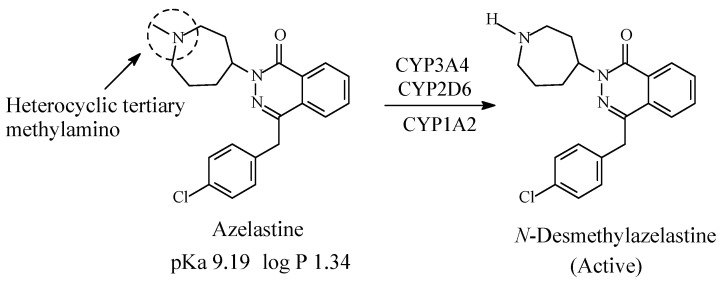
Metabolism of azelastine.

**Figure 24 molecules-26-01917-f024:**
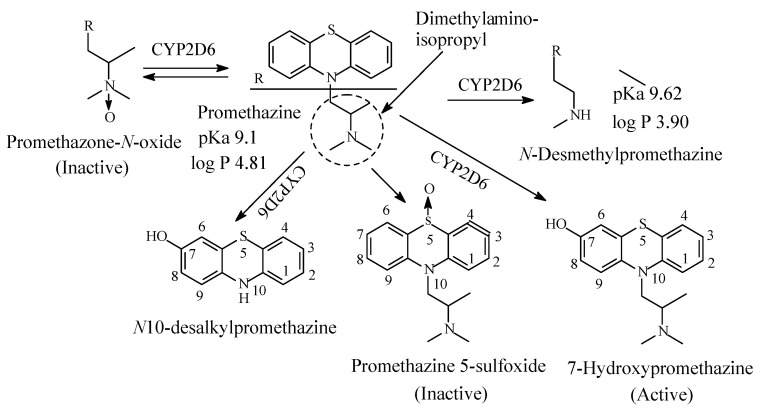
Metabolic pathways of promethazine.

**Figure 25 molecules-26-01917-f025:**
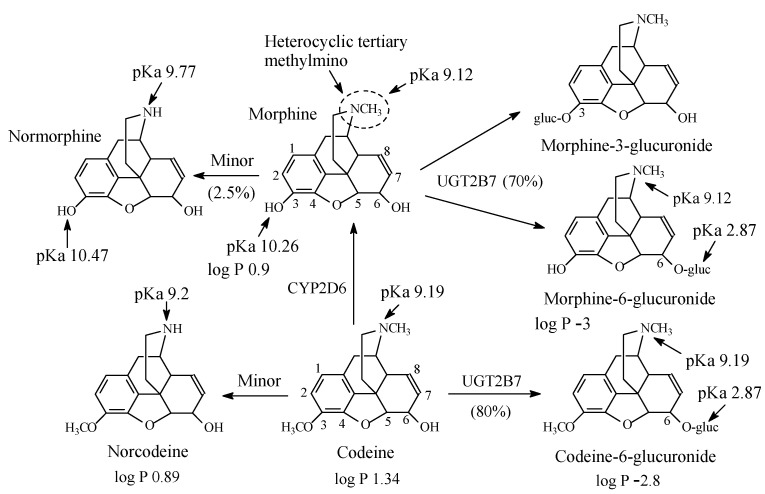
Metabolic pathways of codeine and morphine.

**Figure 26 molecules-26-01917-f026:**
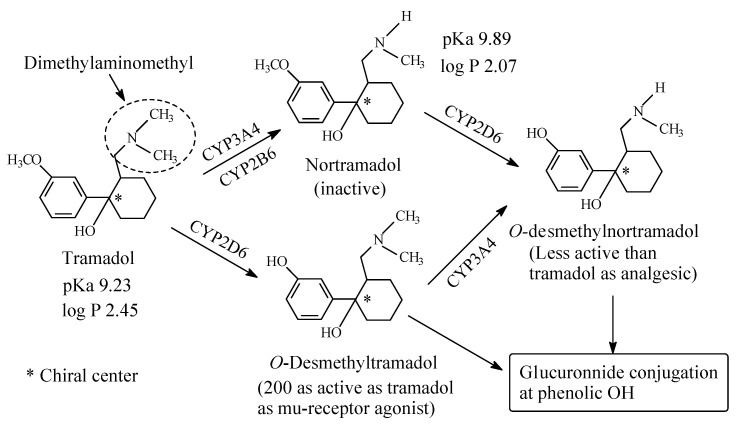
Metabolic pathways of tramadol.

**Figure 27 molecules-26-01917-f027:**
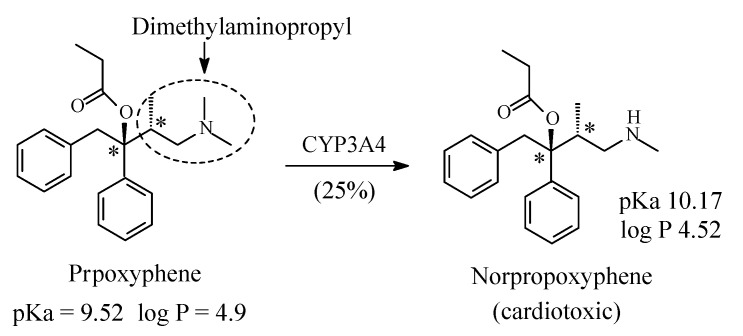
Major metabolic pathway of dextropropoxyphene.

**Figure 28 molecules-26-01917-f028:**
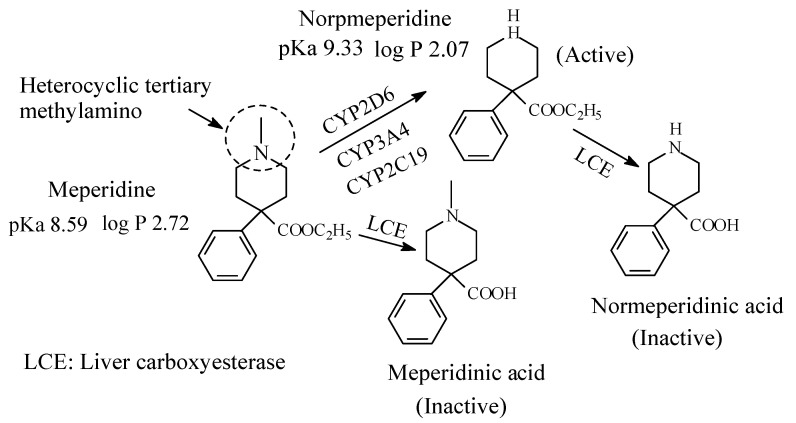
Metabolic pathways of meperidine (pethidine).

**Figure 29 molecules-26-01917-f029:**
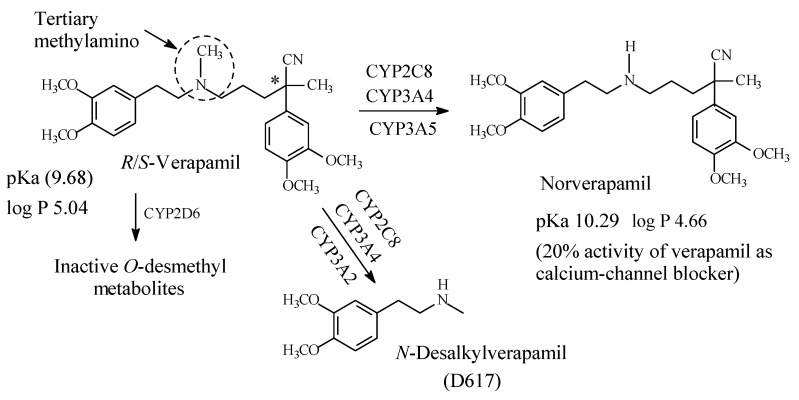
Metabolic pathways of verapamil.

**Figure 30 molecules-26-01917-f030:**
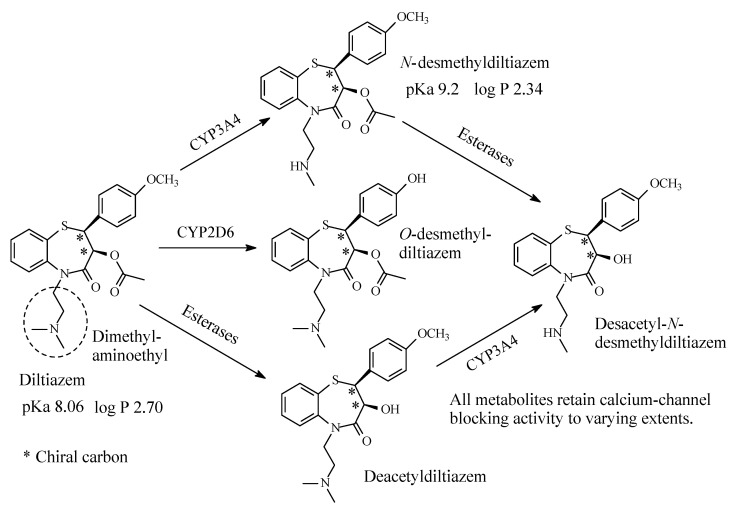
Metabolic pathways of diltiazem.

**Figure 31 molecules-26-01917-f031:**
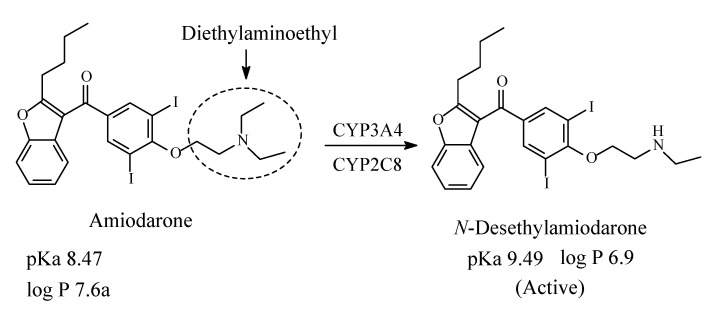
Metabolism of amiodarone.

**Figure 32 molecules-26-01917-f032:**
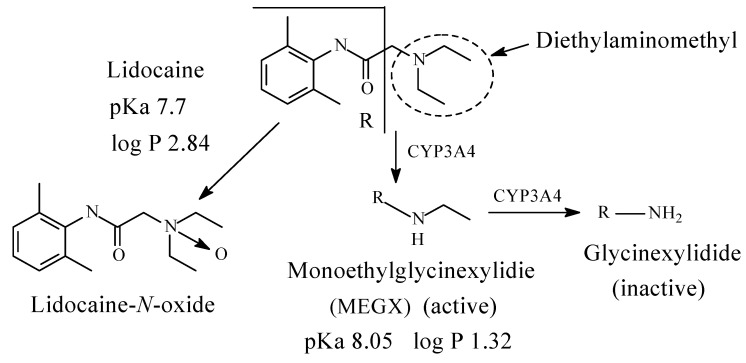
Major metabolic pathways of lidocaine.

**Figure 33 molecules-26-01917-f033:**
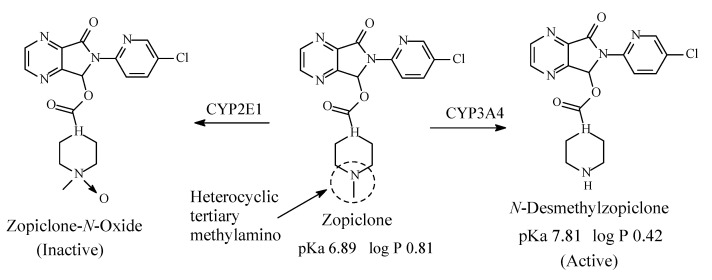
Metabolic pathways of zopiclone.

**Figure 34 molecules-26-01917-f034:**
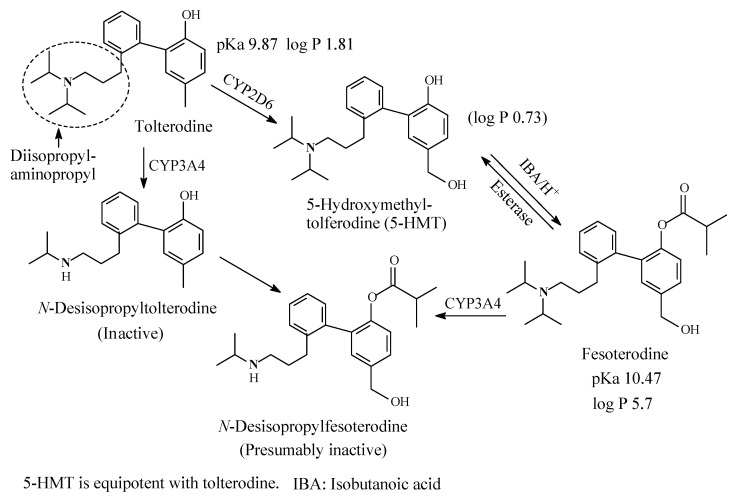
Metabolic pathways of tolterodine and fesoterodine.

**Figure 35 molecules-26-01917-f035:**
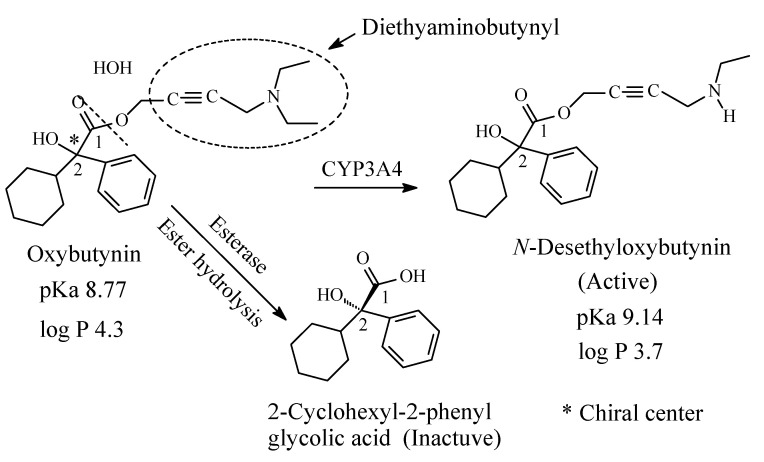
Metabolic pathways of oxybutynin.

**Figure 36 molecules-26-01917-f036:**
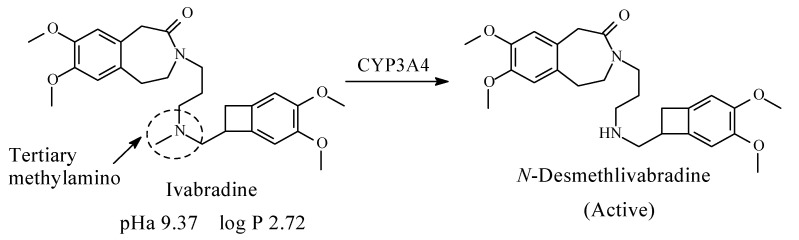
Metabolism of ivabradine.

**Figure 37 molecules-26-01917-f037:**
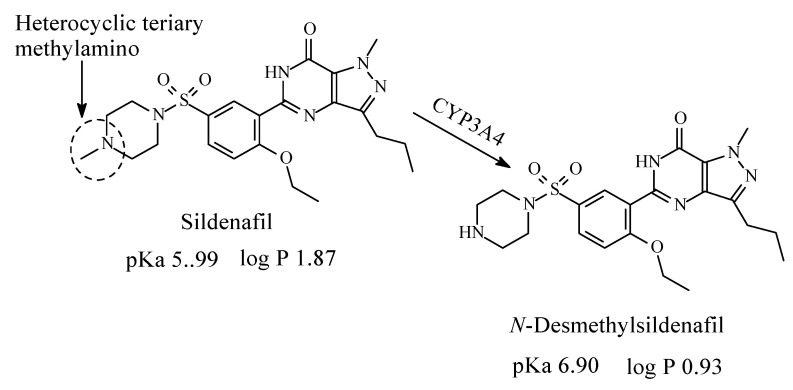
Metabolism of sildenafil.

**Figure 38 molecules-26-01917-f038:**
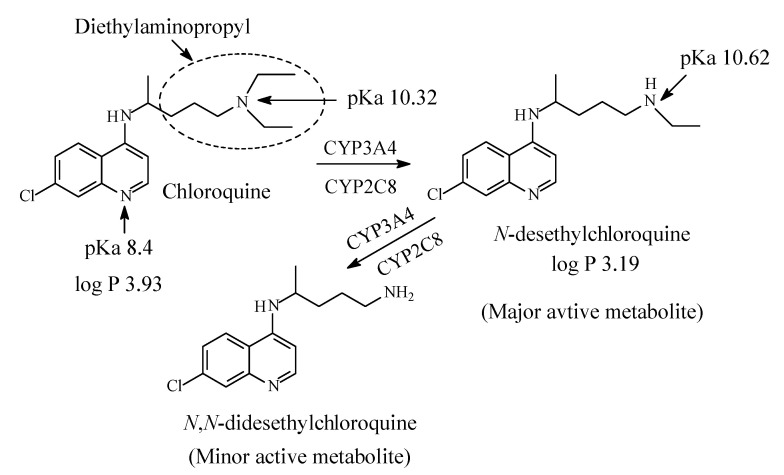
Major metabolic pathways of chloroquine.

**Figure 39 molecules-26-01917-f039:**
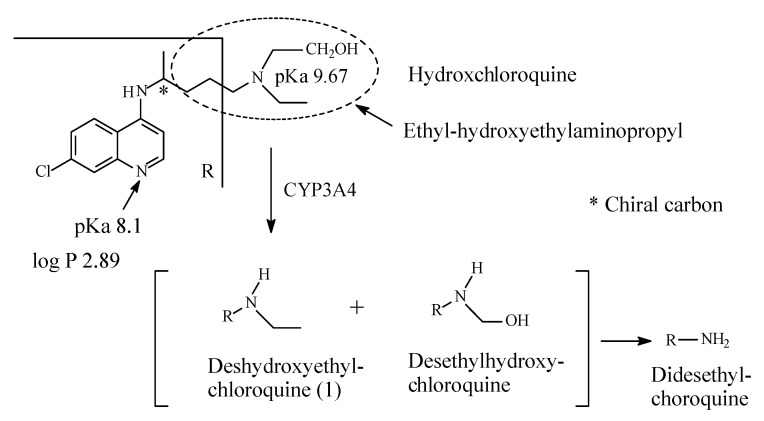
Metabolism of hydroxychloroquine.

**Figure 40 molecules-26-01917-f040:**
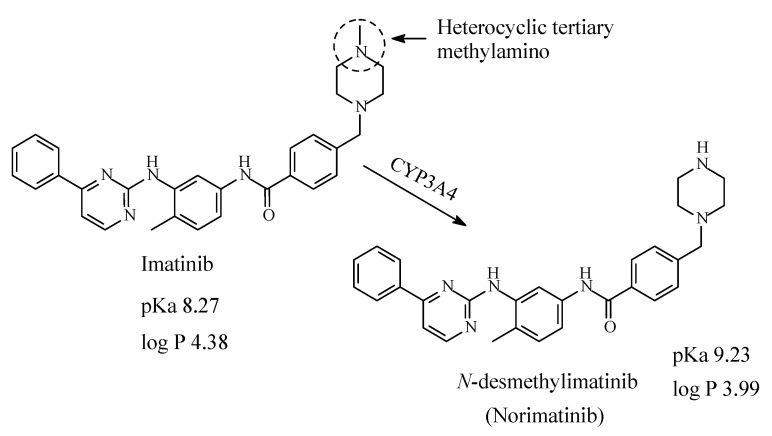
Metabolism of imatinib.

**Figure 41 molecules-26-01917-f041:**
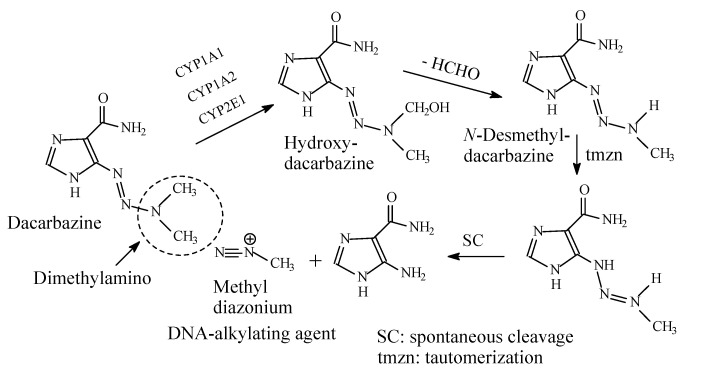
Metabolic activation of dacarbazine.

**Figure 42 molecules-26-01917-f042:**
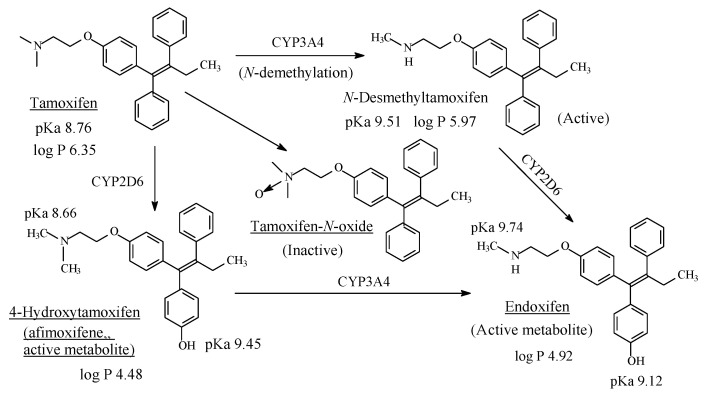
Metabolic pathways of tamoxifen.

**Figure 43 molecules-26-01917-f043:**
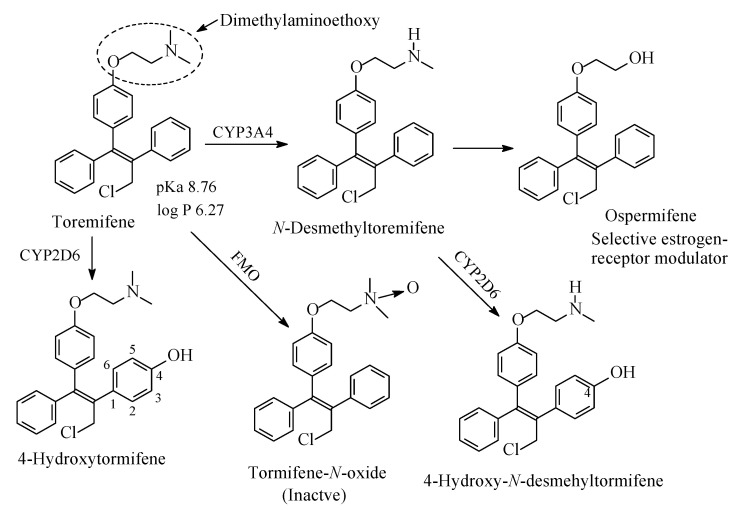
Metabolic pathways of tormifene.

**Figure 44 molecules-26-01917-f044:**
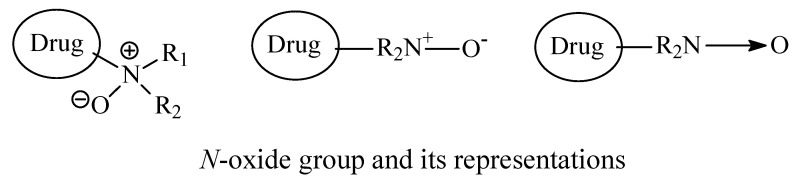
N-oxide group and its representations.

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
