# Peer review of "Metabolic N-Dealkylation and N-Oxidation as Elucidators of the Role of Alkylamino Moieties in Drugs Acting at Various Receptors"

_molecules, 2021, doi:10.3390/molecules26071917_

Round 1

Reviewer 1 Report

This review has addressed the metabolic N-dealkylation and N-oxidation as elucidators of the role of alkylamino moieties in drugs acting at various receptors.

In this manuscript, two aspects were examined after surveying the metabolism of representative alkylamino-moieties-containing drugs that act at various receptors (i) the pharmacologic activities and relevant physicochemical properties of the metabolites with respect to their parent drugs and (ii) the role of alkylamino moieties on the molecular docking of drugs in receptors. N-dealkylation of alkylamino moieties in drugs has resulted in clinically used drugs, potential drugs, activation of prodrugs as well as attenuation and loss of activity of drugs. In contrast, N-oxide metabolites resulting from N-oxidation of dialkylamino moieties are invariably pharmacologically inactive but are bioreducible to the active forms. As thus, they have formed and will form basis of prodrug development.

In conclusion, I have judged that this manuscript is qualified to be good enough for publication in Molecules.

The manuscript is well-written but it seems that there are a few careless mistakes. This paper could be accepted after revision according to the comments described below.

  1. Figure 2, page 5: It seems that the binding style of amino group in ion-ion binding of secondary amine is not correct as the location of one hydrogen atom is inappropriate. It should be described in the same way as that of tertiary amine.

  1. Figure 32, page 26: The position of the dashed line indicating hydrolysis of ester is not appropriate. The dashed line should be written on the ester bond.

  1. Figure 34, page 27: The chemical structures of 8-azapurine scaffold in sildenafil and N-desmethylsildenafil are not correct. The bonds between 5 and 6 positions of 8-azapurine are double, not single.

Author Response

Dear respected Doctor

Re: Molecules 1155844

I thank you so much for reviewing my manuscript, for your positive attitude and for your valuable suggestion of adding dimethylaminoalkyl-moieties-containing phenothiazine drugs. Such inclusion will certainly add to the betterment of the quality of the review.

In response to your suggestion, I have included examples of dimethylaminoalkyl-moieties-containing- phenothiazine drugs in sections 3 and 4 of the review. I hope the additions will be to your satisfaction. 

Reviewer 2 Report

Dear Authors,

Manuscript entitled “Metabolic N-dealkylation and N-oxidation as elucidators of the role of alkylamino moieties in drugs acting at various receptors” is a very extensive material. Writing this manuscript certainly required a lot of work from the author. The author has made an extensive literature review. I missed the mention of phenothiazine neuroleptics in this work. It is a long known and used group of drugs containing dialkolaminoalkyl substituents. I believe that it is worth mentioning this group of compounds in this type of publication.

Moreover, the author should correct minor editing errors

- the authors could standardize the symbol of the methyl substituent, one time it is -CH3 and another time it is only – (only binding), both forms are correct and look good, but within one article it should be the same

- the names of the figures in the lines 184, 239, 439 are written in larger type

- there were no spaces in the lines: 194, 197, 282, 331, 503

- lines 640 - 648 are in larger font

The author should also format the references more carefully.

Author Response

Dear respected Doctor

Re: Molecules 1155844

I thank you so much for reviewing my manuscript, for your positive attitude and for the meticulous corrections to be made in the figures.

I have made the corrections in figures 2, 32 and 34 in response to your recommendation hoping you will find them satisfactory.